# PREDFORMER: TRANSFORMERS ARE EFFECTIVE SPATIAL-TEMPORAL PREDICTIVE LEARNERS

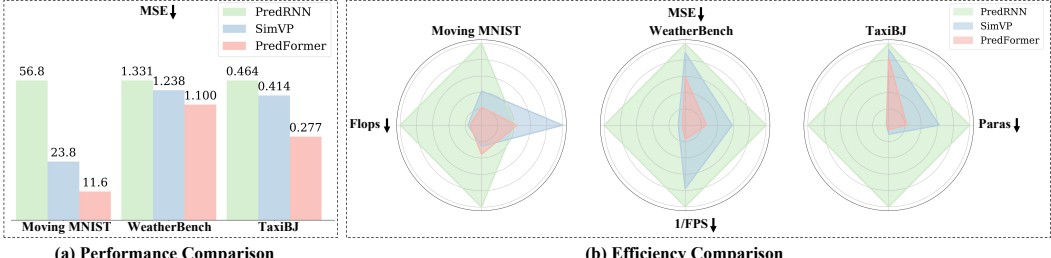

Figure 1: (a) Performance of PredRNN, SimVP, and PredFormer; (b) Model efficiency comparison. The more inside model indicates better accuracy and efficiency.

## ABSTRACT

Spatiotemporal predictive learning methods generally fall into two categories: recurrent-based approaches, which face challenges in parallelization and performance, and recurrent-free methods, which employ convolutional neural networks (CNNs) as encoder-decoder architectures. These methods benefit from strong inductive biases but often at the expense of scalability and generalization. This paper proposes **PredFormer**, a pure transformer-based framework for spatiotemporal predictive learning. Motivated by the Vision Transformers (ViT) design, PredFormer leverages carefully designed Gated Transformer blocks, following a comprehensive analysis of 3D attention mechanisms, including full-, factorized-, and interleaved- spatial-temporal attention. With its recurrent-free, transformer-based design, PredFormer is both simple and efficient, significantly outperforming previous methods by large margins. Extensive experiments on synthetic and real-world datasets demonstrate that PredFormer achieves state-of-the-art performance. On Moving MNIST, PredFormer achieves a 51.3% reduction in MSE relative to SimVP. For TaxiBJ, the model decreases MSE by 33.1% and boosts FPS from 533 to 2364. Additionally, on WeatherBench, it reduces MSE by 11.1% while enhancing FPS from 196 to 404. These performance gains in both accuracy and efficiency demonstrate PredFormer's potential for real-world applications. The source code and trained models will be made available to the public.

## 1 INTRODUCTION

Spatio-temporal predictive learning involves learning spatial and temporal patterns by predicting future frames based on past observations. This capability is essential for various applications, including weather forecasting (Rasp et al., 2020; Pathak et al., 2022; Bi et al., 2023), traffic flow prediction (Fang et al., 2019; Wang et al., 2019), precipitation nowcasting (Shi et al., 2015; Gao et al., 2022b) and human motion forecasting (Zhang et al., 2017b; Wang et al., 2018a).

Despite the success of various spatial-temporal prediction learning methods, they often struggle to balance computation cost and performance. On the one hand, high-powered recurrent-based methods (Shi et al., 2015; Wang et al., 2017; 2019; Chang et al., 2021; Yu et al., 2019; Tang et al., 2023; 2024) rely heavily on autoregressive RNN frameworks, which face significant limitations in parallelization and computational efficiency. On the other hand, efficient recurrent-free methods (Gao

et al., 2022a; Tan et al., 2023a), such as those based on the SimVP framework, employ CNNs within an encoder-decoder architecture but are constrained by the local receptive field, limiting their scalability and generalization. This raises a more fundamental question: *Can we develop a framework that autonomously learns spatiotemporal dependencies without relying on inductive bias?*

An intuitive solution directly adopts a pure transformer (Vaswani et al., 2017) structure, as it is an efficient alternative to RNNs and has better scalability than CNNs. Transformers have demonstrated remarkable success in visual tasks (Dosovitskiy et al., 2020; Liu et al., 2021; Bertasius et al., 2021; Arnab et al., 2021; Tarasiou et al., 2023). Previous methods try to combine Swin Transformer (Liu et al., 2021) in recurrent-based frameworks such as SwinLSTM (Tang et al., 2023) and integrate MetaFormer (Yu et al., 2022) as a temporal translator in recurrent-free CNN-based encoder-decoder frameworks such as OpenSTL (Tan et al., 2023b). Despite these advances, pure transformer-based architecture remains underexplored mainly due to the challenges of capturing spatial and temporal relationships within a unified framework. While merging spatial and temporal dimensions and applying full attention is conceptually straightforward, it is computationally expensive due to the quadratic scaling of attention with sequence length. Several recent methods (Bertasius et al., 2021; Arnab et al., 2021; Tarasiou et al., 2023) decouple full attention and show that spatial and temporal relations can either be treated separately in a factorized or interleaved manner to reduce complexity.

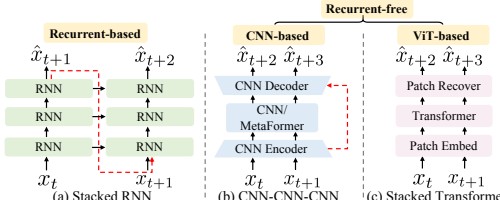

Figure 2: Main categories of spatiotemporal predictive learning framework. (a) Recurrent-based Framework (b) CNN Encoder-Decoder-based Recurrent-free Framework. (c) Pure transformer-based Recurrent-free Framework.

In this work, we propose PredFormer, a pure transformer-based architecture for spatiotemporal predictive learning. PredFormer dives into the decomposition of spatial and temporal transformers, integrating self-attention with gated linear units (Dauphin et al., 2017) to more effectively capture complex spatiotemporal dynamics. In addition to retaining spatial-temporal full attention encoder and factorized encoder strategies for both spatial-first and temporal-first configurations, we introduce six novel interleaved spatiotemporal transformer architectures, resulting in nine configurations. This exploration is motivated by the varying spatial and temporal resolutions and dependencies across different tasks and datasets. This comprehensive investigation pushes the boundaries of current models and sets valuable benchmarks for spatial-temporal modeling.

Notably, PredFormer achieves state-of-the-art performance across three benchmark datasets, including synthetic moving object prediction, traffic flow prediction, and weather forecasting, outperforming previous methods by a substantial margin without relying on complex model architectures or specialized loss functions. Moreover, our optimal model excels in performance and is efficient, offering fewer parameters, lower FLOPs, and faster inference speeds than previous models. This highlights its strong potential for real-world applications.

The main contributions can be summarized as follows:

- We propose PredFormer, a pure gated transformer-based model for spatiotemporal predictive learning. By eliminating the inductive biases inherent in CNNs, PredFormer harnesses transformers' scalability and generalization capabilities, positioning it as a highly adaptable model with significantly enhanced potential and performance ceilings.

- We perform an in-depth analysis of spatial-temporal transformer factorization, exploring full attention encoders and factorized encoders along with interleaved spatiotemporal transformer architectures, resulting in nine PredFormer variants. These variants address the differing spatial and temporal resolutions across tasks and datasets for optimal performance.

- We conduct a comprehensive study on training ViT from scratch on small datasets, exploring regularization and position encoding techniques.

- Extensive experiments demonstrate PredFormer's outstanding performance. Compared to SimVP, on Moving MNIST it reduces MSE by 51.3%, on TaxiBJ by 33.1% while increasing FPS from 533 to 2364, and on WeatherBench by 11.1% with FPS rising from 196 to

404. These results highlight PredFormer's superior accuracy and efficiency and emphasize its potential for real-world applications. We will release our code and trained models.

## 2 RELATED WORK

**Recurrent-based spatial-temporal predictive learning.** Recent advancements in recurrent-based spatiotemporal predictive models have integrated CNNs, ViTs, and Vision Mamba (Liu et al., 2024) into RNNs, employing various strategies to capture spatiotemporal relationships. ConvLSTM (Shi et al., 2015), evolving from FC-LSTM (Srivastava et al., 2015), innovatively integrates convolutional operations into the LSTM framework. PredNet (Lotter et al., 2017) leverages deep recurrent convolutional neural networks with bottom-up and top-down connections to predict future video frames. PredRNN (Wang et al., 2017) introduces the Spatiotemporal LSTM (ST-LSTM) unit, which effectively captures and memorizes spatial and temporal representations by propagating hidden states horizontally and vertically. PredRNN++ (Wang et al., 2018b) incorporates a gradient highway unit and Causal LSTM to address the vanishing gradient problem and adaptively capture temporal dependencies. E3D-LSTM (Wang et al., 2018c) extends the memory capabilities of ST-LSTM by integrating 3D convolutions. The MIM model (Wang et al., 2019) further refines the ST-LSTM by reimagining the forget gate with dual recurrent units and utilizing differential information between hidden states. CrevNet (Yu et al., 2019) employs a CNN-based reversible architecture to decode complex spatiotemporal patterns efficiently. PredRNNv2 (Wang et al., 2022) enhances PredRNN by introducing a memory decoupling loss and a curriculum learning strategy. MAU (Chang et al., 2021) adds a motion-aware unit specifically designed to capture dynamic motion information. SwinLSTM (Tang et al., 2023) advances spatiotemporal modeling by integrating the Swin Transformer (Liu et al., 2021) module into the LSTM architecture, while VMRNN (Tang et al., 2024) extend this by incorporating the Vision Mamba module. Unlike these approaches, PredFormer is a recurrent-free method offering superior efficiency.

**Recurrent-free spatial-temporal predictive learning.** Recent recurrent-free models, e.g., SimVP (Gao et al., 2022a), are developed based on a CNN-based encoder-decoder with a temporal translator. TAU (Tan et al., 2023a) builds upon this by separating temporal attention into static intra-frame and dynamic inter-frame components, introducing a differential divergence loss to supervise inter-frame variations. OpenSTL (Tan et al., 2023b) integrates a MetaFormer model as the temporal translator. Additionally, PhyDNet (Guen & Thome, 2020) incorporates physical principles into CNN architectures, while DMVFN (Hu et al., 2023) introduces a dynamic multi-scale voxel flow network to enhance video prediction performance. EarthFormer (Gao et al., 2022b) presents a 2D CNN encoder-decoder architecture with cuboid attention. WAST (Nie et al., 2024) proposes a wavelet-based method, coupled with a wavelet-domain High-Frequency Focal Loss. In contrast to prior methods, PredFormer advances spatiotemporal learning with its recurrent-free, pure transformer-based architecture, leveraging a global receptive field to achieve superior performance, outperforming prior models without relying on complex architecture designs or specialized loss.

**Vision Transformer (ViT).** ViT (Dosovitskiy et al., 2020) has demonstrated exceptional performance across various vision tasks. In the field of video processing, TimeSformer (Bertasius et al., 2021) investigates the factorization of spatial and temporal self-attention and proposes that divided attention where temporal and spatial attention are applied separately yields the best accuracy. ViViT (Arnab et al., 2021) explores factorized encoders, self-attention, and dot-product mechanisms, concluding that a factorized encoder with spatial attention applied first performs better. On the other hand, TSViT (Tarasiou et al., 2023) finds that a factorized encoder prioritizing temporal attention achieves superior results. Latte (Ma et al., 2024) investigates factorized encoders and factorized self-attention mechanisms, incorporating both spatial-first and spatial-temporal parallel designs, within the context of latent diffusion transformers for video generation. Despite these advancements, most existing models primarily focus on video classification, with limited research on applying ViTs to spatiotemporal predictive learning. Moving beyond earlier methods that focus on factorizing self-attention, PredFormer explores the decomposition of spatial and temporal transformers at a deeper level by integrating self-attention with gated linear units and introducing innovative interleaved designs, allowing for a more robust capture of complex spatiotemporal dynamics.

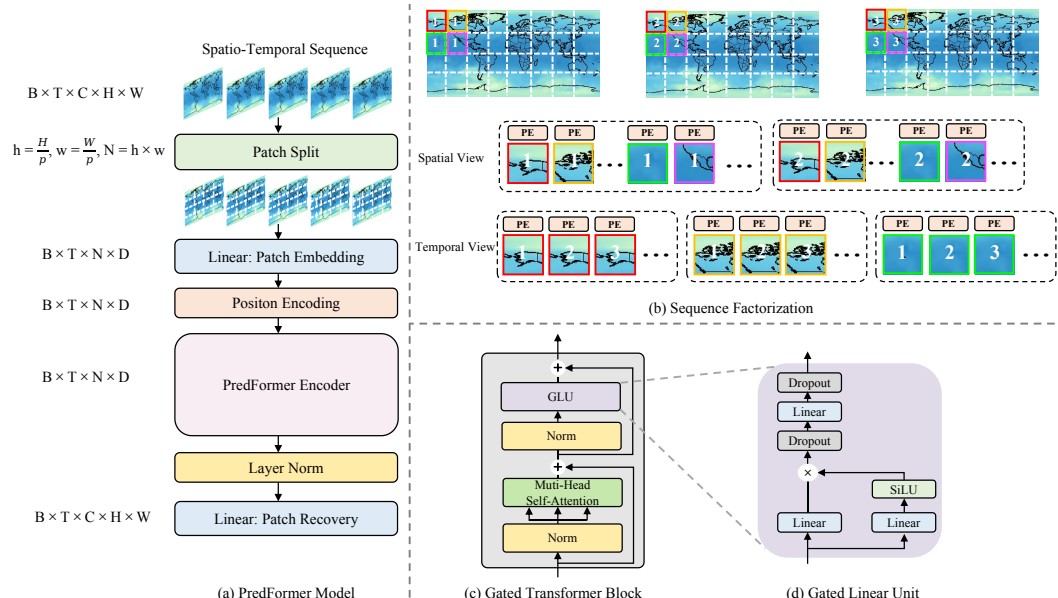

Figure 3: (a) Overview of the PredFormer model framework. (b) Sequence factorization from spatial view and temporal view. (c) Gated Transformer Block. (d) Gated Linear Unit.

## 3 METHOD

To systematically analyze the transformer structure of the network model in spatial-temporal predictive learning, we propose the PredFormer as a general model design, as shown in Fig 3(a).

In the following sections, we first introduce the pure transformer-based architecture in Sec 3.1. Next, we describe the Gated Transformer Block (GTB) in Sec 3.2. Finally, we present how to use GTB to build a PredFormer layer and architecture variants in Sec 3.3.

### 3.1 PURE TRANSFORMER BASED ARCHITECTURE

**Patch Embedding.** Follow the ViT design, PredFormer splits a sequence of frames $\mathcal{X}$ into a sequence of $N = \left\lfloor \frac{H}{p} \right\rfloor \left\lfloor \frac{W}{p} \right\rfloor$ equally sized, non-overlapping patches of size $p$, each of which is flattened into a 1D tokens. These tokens are then linearly projected into hidden dimensions $D$ and processed by a layer normalization (LN) layer, resulting in a tensor $\mathcal{X}' \in \mathbb{R}^{B \times T \times N \times D}$.

**Position Encoding.** Unlike the typical ViT approach, which employs learnable position embeddings, we incorporate a 2D spatiotemporal position encoding (PE) generated by sinusoidal functions with absolute coordinates for each patch.

**PredFormer Encoder.** The 1D tokens are then processed by a PredFormer Encoder for feature extraction. PredFormer Encoder is stacked by Gated Transformer Blocks in various manners.

**Patch Recovery.** Since our encoder is based on a pure gated transformer, without convolution or resolution reduction, global context is modeled at every layer. This allows it to be paired with a simple decoder, forming a powerful prediction model. After the encoder, a linear layer serves as the decoder, projecting the hidden dimensions back to recover the 1D tokens to 2D patches.

### 3.2 GATED TRANSFORMER BLOCK

The Standard Transformer model (Vaswani et al., 2017) alternates between Multi-Head Attention (MSA) and Feed-Forward Networks (FFN). The attention mechanism for each head is defined as:

$$\text{Attention}(\mathbf{Q}, \mathbf{K}, \mathbf{V}) = \text{Softmax}\left(\frac{\mathbf{Q}\mathbf{K}^\top}{\sqrt{d_k}}\right)\mathbf{V}, \qquad (1)$$

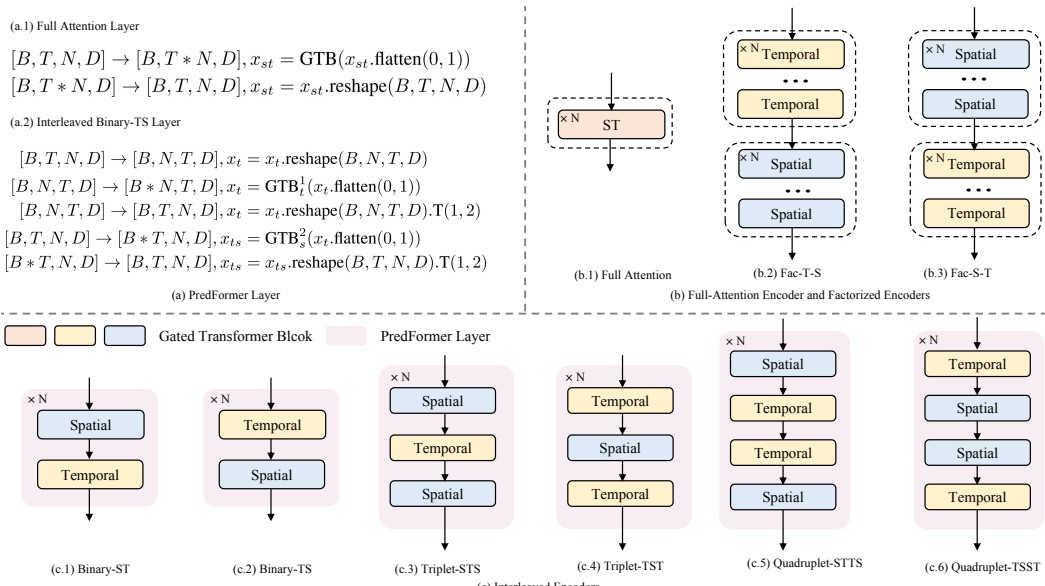

Figure 4: (a) Data transform of Full Attention layer and Binary-TS layer (b) Full Attention Encoder and Factorized Encoders (c) Interleaved Encoders with Binary, Triplet, and Quadrupled design

where in self-attention, the queries $\mathbf{Q}$, keys $\mathbf{K}$, and values $\mathbf{V}$ are linear projections of the input $\mathbf{X}$, represented as $\mathbf{Q} = \mathbf{X}\mathbf{W}_q$, $\mathbf{K} = \mathbf{X}\mathbf{W}_k$, and $\mathbf{V} = \mathbf{X}\mathbf{W}_v$, with $\mathbf{X}, \mathbf{Q}, \mathbf{K}, \mathbf{V} \in \mathbb{R}^{N \times d}$. The FFN then processes each position in the sequence by applying two linear transformations.

Gated Linear Units (GLUs) (Dauphin et al., 2017), often used in place of simple linear transformations, involve the element-wise product of two linear projections, with one projection passing through a sigmoid function. Various GLU variants control the flow of information by substituting the sigmoid with other non-linear functions. For instance, SwiGLU (Shazeer, 2020) replaces the sigmoid with the Swish activation function (SiLU) (Hendrycks & Gimpel, 2016), as shown in Eq 2.

$$\text{Swish}_\beta(x) = x\sigma(\beta x)$$
$$\text{SwiGLU}(x, W, V, b, c, \beta) = \text{Swish}_\beta(xW + b) \otimes (xV + c) \tag{2}$$

SwiGLU has been demonstrated to outperform Multi-layer Perceptrons (MLPs) in various natural language processing tasks(Shazeer, 2020). Inspired by the SwiGLU's success in these tasks, our Gated Transformer Block (GTB), shown in Fig 3(c), incorporates MSA followed by a SwiGLU-based FFN, as illustrated in Fig 3(d). GTB is defined as:

$$\mathbf{Y}^l = \text{MSA}(\text{LN}(\mathbf{Z}^l)) + \mathbf{Z}^l$$
$$\mathbf{Z}^{l+1} = \text{SwiGLU}(\text{LN}(\mathbf{Y}^l)) + \mathbf{Y}^l \tag{3}$$

### 3.3 VARIANTS OF PREDFORMER

Modeling spatiotemporal dependencies in predictive learning is challenging, as the balance between spatial and temporal information differs across tasks and datasets. This requires flexible, adaptive models that can accommodate varying dependencies and scales. To address these, we explore both full-attention encoders and factorized encoders with spatial-first (Fac-S-T) and temporal-first (Fac-T-S) configurations, as shown in Fig 4(b). In addition, we introduce six interleaved models based on PredFormer layer, enabling dynamic interaction across multiple scales.

A PredFormer layer is a module capable of simultaneously processing spatial and temporal information. Building on this design principle, we propose **three** interleaved spatiotemporal paradigms, Binary, Triplet, and Quadrupled, which sequentially model the relation from spatial view and temporal view as depicted in Fig 3(c). Ultimately, they yield **six** distinct architectural configurations. A detailed illustration of these nine variants is provided in Fig 4.

Table 1: Benchmark datasets for PredFormer.

| Dataset | Training size | Testing size | Channel | Height | Width | $T$ | $T'$ | Interval |
|---|---|---|---|---|---|---|---|---|
| Moving MNIST | 10,000 | 10,000 | 1 | 64 | 64 | 10 | 10 | - |
| WeatherBench-S | 52559 | 17495 | 1 | 32 | 64 | 12 | 12 | 30 min |
| TaxiBJ | 20,461 | 500 | 2 | 32 | 32 | 4 | 4 | 1 hour |

For full attention layers, given input $\mathcal{X} \in \mathbb{R}^{B \times T \times N \times D}$, attention is computed over the sequence of length $T \times N$. As illustrated in Fig 4 (a.1) and (b.1), we merge and flatten the spatial and temporal tokens to compute attention through several stacked $\text{GTB}_{st}$.

For Binary layers, each GTB block processes temporal or spatial sequence independently, where we denote Binary-TS or Binary-ST layer, as illustrated in Fig 4 (a.2) and (c.2). The input is first reshaped, and processed through $\text{GTB}_t^1$, where attention is applied over the temporal sequence. The tensor is then reshaped back to restore the temporal order. Subsequently, spatial attention is applied using another $\text{GTB}_s^2$, where the tensor is flattened along the temporal dimension and processed.

For Triplet and Quadruplet layers, additional blocks are stacked on top of the Binary structure. Triplet-TST captures more temporal dependencies, while Triplet-STS focuses more on spatial dependencies, both using the same number of parameters. The Quadruplet layer combines two Binary layers in different orders. We omit further detailed explanations.

## 4 EXPERIMENTS

We present extensive evaluations of PredFormer and state-of-the-art models. We conduct experiments across synthetic and real-world scenarios, including long-term prediction(moving object trajectory prediction and weather forecasting), and short-term prediction(traffic flow prediction). The dataset statistics are presented in Tab 1. These datasets have different spatial resolutions, temporal frames, and intervals, which determine their different spatiotemporal dependencies.

**Implementation Details** Our method is implemented in PyTorch, with experiments conducted on 24GB NVIDIA RTX 3090 and 24GB A5000 GPUs, unless otherwise specified. All experiments are run with a single GPU. PredFormer is optimized using the AdamW (Loshchilov & Hutter, 2019) optimizer with an L2 loss, a weight decay of 1e-2, and a learning rate selected from {5e-4, 1e-3} for best performance. OneCycle scheduler is used for Moving MNIST and TaxiBJ, while the Cosine scheduler is applied for WeatherBench. Dropout (Hinton, 2012) and stochastic depth (Huang et al., 2016) regularization prevent overfitting for TaxiBJ and WeatherBench. Further hyperparameters details are provided in Appendix Sec A.2. For different PredFormer variants, we maintain a constant number of GTB blocks to ensure comparable parameters. In cases where the Triplet model cannot be evenly divided, we use the number of GTB blocks closest to the others.

**Evaluation Metrics** We assess model performance using a suite of metrics across three dimensions. (1) **Pixel-wise error** is measured using Mean Squared Error (MSE), Mean Absolute Error (MAE), and Root Mean Squared Error (RMSE). (2) **Predicted frame quality** is evaluated through similarity metrics Structural Similarity Index Measure (SSIM) (Wang et al., 2004). Lower values of MSE, MAE, and RMSE, combined with higher SSIM, signify better predictions. (3) **Computational efficiency** is assessed by the number of parameters, floating-point operations (FLOPs), and inference speed in frames per second (FPS) on an NVIDIA A5000 GPU. This multi-faceted evaluation framework comprehensively evaluates the model's accuracy, efficiency, and scalability.

### 4.1 LONG-TERM PREDICTION: MOVING MNIST

**Moving MNIST.** The moving MNIST dataset (Srivastava et al., 2015) serves as a benchmark synthetic dataset for evaluating sequence prediction models. We follow (Srivastava et al., 2015) to generate Moving MNIST sequences with 20 frames, using the initial 10 frames for input and the subsequent 10 frames as the prediction target. We adopt 10000 sequences for training and for fair comparisons, we use the pre-generated 10000 sequences (Gao et al., 2022a) for validation.

Table 2: Quantitative comparison on **Moving MNIST**. Each model observes 10 frames and predicts the subsequent 10 frames. We train our models for **200** epochs and cite other results of the original paper.

| Method | Paras(M) | Flops(G) | FPS | MSE ↓ | MAE ↓ | SSIM ↑ |
|---|---|---|---|---|---|---|
| **Recurrent-free** | | | | | | |
| SimVP | 58.0 | 19.4 | 209 | 32.2 | 89.1 | 0.927 |
| TAU | 44.7 | 16.0 | 283 | 24.6 | 71.9 | 0.945 |
| **PredFormer-ps4** | | | | | | |
| Full Attention | 25.3 | 145.0 | 13 | 26.4 | 76.7 | 0.941 |
| Fac-S-T | 25.3 | 68.9 | 52 | 35.8 | 95.3 | 0.920 |
| Fac-T-S | 25.3 | 68.8 | 53 | 24.3 | 70.6 | 0.946 |
| Binary-TS | 25.3 | 68.9 | 63 | 20.7 | 63.7 | **0.955** |
| Binary-ST | 25.3 | 68.9 | 65 | 20.6 | 63.2 | **0.955** |
| **Triplet-TST** | 25.3 | 67.6 | 69 | **20.5** | **63.1** | **0.955** |
| Triplet-STS | 25.3 | 70.2 | 65 | 20.7 | 63.8 | 0.953 |
| Quadruplet-TSST | 25.3 | 68.9 | 52 | 20.7 | 63.6 | **0.955** |
| Quadruplet-STTS | 25.3 | 68.9 | 50 | 20.9 | 64.4 | 0.954 |
| **PredFormer-ps8** | | | | | | |
| Full Attention | 25.3 | 21.2 | 120 | 30.2 | 86.3 | 0.932 |
| Fac-S-T | 25.3 | 16.5 | 161 | 43.5 | 113.5 | 0.899 |
| Fac-T-S | 25.3 | 16.5 | 163 | 31.2 | 88.3 | 0.929 |
| Binary-TS | 25.3 | 16.5 | 119 | 27.3 | 80.6 | 0.938 |
| Binary-ST | 25.3 | 16.5 | 148 | 27.8 | 80.5 | 0.937 |
| Triplet-TST | 25.3 | 16.4 | 148 | 26.9 | 78.8 | 0.939 |
| Triplet-STS | 25.3 | 16.5 | 159 | 29.1 | 84.5 | 0.933 |
| **Quadruplet-TSST** | 25.3 | 16.5 | 148 | **26.0** | **77.2** | **0.941** |
| Quadruplet-STTS | 25.3 | 16.5 | 154 | 29.0 | 84.5 | 0.933 |

Table 3: Quantitative comparison on **Moving MNIST**. Each model observes 10 frames and predicts the subsequent ten frames. We train our models for **2000** epochs and cite other results of the original paper.

| Method | Paras(M) | Flops(G) | FPS | MSE ↓ | MAE ↓ | SSIM ↑ |
|---|---|---|---|---|---|---|
| **Recurrent-based** | | | | | | |
| ConvLSTM | 15.0 | 56.8 | 113 | 103.3 | 182.9 | 0.707 |
| PredRNN | 23.8 | 116.0 | 54 | 56.8 | 126.1 | 0.867 |
| PredRNN++ | 38.6 | 171.7 | 38 | 46.5 | 106.8 | 0.898 |
| MIM | 38.0 | 179.2 | 37 | 44.2 | 101.1 | 0.910 |
| E3D-LSTM | 51.0 | 298.9 | 18 | 41.3 | 86.4 | 0.910 |
| PhyDNet | 3.1 | 15.3 | 182 | 24.4 | 70.3 | 0.947 |
| MAU | 4.5 | 17.8 | 201 | 27.6 | 86.5 | 0.937 |
| PredRNNv2 | 24.6 | 708.0 | 24 | 48.4 | 129.8 | 0.891 |
| **Recurrent-free** | | | | | | |
| SimVP | 58.0 | 19.4 | 209 | 23.8 | 68.9 | 0.948 |
| TAU | 44.7 | 16.0 | 283 | 19.8 | 60.3 | 0.957 |
| **PredFormer-ps4** | | | | | | |
| Triplet-TST | 25.3 | 67.6 | 110 | 11.9 | 42.0 | 0.974 |
| **Triplet-STS** | 25.3 | 70.2 | 93 | **11.6** | **41.4** | **0.975** |
| **PredFormer-ps8** | | | | | | |
| Fac-T-S | 25.3 | 16.5 | 170 | 16.9 | 55.8 | 0.963 |
| Binary-TS | 25.3 | 16.5 | 147 | 12.8 | 46.1 | 0.972 |
| Triplet-TST | 25.3 | 16.4 | 165 | 13.4 | 47.2 | 0.971 |
| **Quadruplet-TSST** | 25.3 | 16.5 | 152 | **12.5** | **44.6** | **0.973** |

On the Moving MNIST dataset, the most commonly used benchmark dataset, we employ two training settings to explore the performance, convergence, efficiency, and variants of our PredFormer framework. In the first setting, we train 200 epochs to compare the performance of our nine proposed models with SimVP and TAU, we present our quantitative results in Tab 2. In the second setting, following previous work (Gao et al., 2022a; Tan et al., 2023a), we train our best-performing models from the 200-epoch runs for 2000 epochs, reporting the final results in Tab 3. We cite the results of all other methods from each original paper for a fair comparison.

**Can PredFormer Converge Faster than SimVP?** When using a patch size of 4, our six interleaved models trained for only 200 epochs surpass the 2000-epoch performance of SimVP (MSE 23.8). This demonstrates that PredFormer achieves faster convergence compared to SimVP. The model's ability to converge in limited epochs while maintaining superior performance highlights the efficiency and robustness of the pure ViT framework over CNN-based approaches.

**Upper Bound Comparison between ViT and CNN framework.** Extending the training of our best-performing 200-epoch model with patch size 4, Triplet-STS (MSE 20.7), to 2000 epochs resulted in a dramatic reduction in MSE to 11.6. This marks a 51.3% improvement over SimVP and a 41.4% improvement over TAU. These results confirm that our pure transformer-based model outperforms all previous methods by a large margin. While CNNs are constrained by inductive bias, they struggle to match the global receptive field advantages of pure transformer architectures, further emphasizing the superior upper bound of PredFormer in spatiotemporal modeling.

**Accuracy and Efficiency Trade-off.** With a patch size of 4, despite having fewer parameters than SimVP, PredFormer has higher FLOPs and lower FPS. We increase the patch size to 8 to balance performance and efficiency, reducing computation to a quarter of the original. In this configuration, FLOPs drop to 16.4G, lower than SimVP's 19.4G and comparable to TAU's 16.0G, with FPS slightly lower than SimVP. When training for 200 epochs, the MSE of PredFormer is higher than SimVP but lower than TAU's 200-epoch results. After extending the training to 2000 epochs, SimVP's MSE improves from 32.2 to 23.8, TAU improves from 24.6 to 19.8, while our PredFormer shows a greater improvement from 26.0 to 12.5. This again demonstrates the higher upper bound of the pure transformer model compared to CNN even with a larger patch size. Specifically, PredFormer achieved a 47.5% improvement over SimVP and a 36.9% improvement over TAU, realizing an impressive accuracy-efficiency trade-off with significant performance gains.

Table 4: Quantitative comparison on **Weather-Bench(T2m)**. Each model observes 12 frames and predicts the subsequent 12 frames. We cite other results from (Tan et al., 2023b).

Table 5: Quantitative comparison on **TaxiBJ**. Each model observes 4 frames and predicts the subsequent 4 frames. We cite other results of the original paper.

| Method | Paras(M) | Flops(G) | FPS | MSE ↓ | MAE ↓ | RMSE ↓ |
|---|---|---|---|---|---|---|
| **Recurrent-based** | | | | | | |
| ConvLSTM | 14.9 | 136.0 | 46 | 1.521 | 0.7949 | 1.233 |
| PredRNN | 23.6 | 278.0 | 22 | 1.331 | 0.7246 | 1.154 |
| PredRNN++ | 38.3 | 413 | 15 | 1.634 | 0.7883 | 1.278 |
| MIM | 37.8 | 109.0 | 126 | 1.784 | 0.8716 | 1.336 |
| PhyDNet | 3.1 | 36.8 | 177 | 285.9 | 8.7370 | 16.91 |
| MAU | 5.5 | 39.6 | 237 | 1.251 | 0.7036 | 1.119 |
| PredRNNv2 | 23.6 | 279.0 | 22 | 1.545 | 0.7986 | 1.243 |
| **Recurrent-free** | | | | | | |
| SimVP | 14.8 | 8.0 | 196 | 1.238 | 0.7037 | 1.113 |
| TAU | 12.2 | 6.7 | 229 | 1.162 | 0.6707 | 1.078 |
| **PredFormer** | | | | | | |
| Full Attention | 5.3 | 17.8 | 101 | 1.126 | 0.6540 | 1.061 |
| Fac-S-T | 5.3 | 8.5 | 431 | 1.783 | 0.8688 | 1.335 |
| **Fac-T-S** | 5.3 | 8.5 | 404 | **1.100** | **0.6469** | **1.049** |
| Binary-TS | 5.3 | 8.6 | 376 | 1.115 | 0.6508 | 1.056 |
| Binary-ST | 5.3 | 8.6 | 397 | 1.140 | 0.6571 | 1.068 |
| Triplet-TST | 4.0 | 6.3 | 521 | 1.108 | 0.6492 | 1.053 |
| Triplet-STS | 4.0 | 6.5 | 530 | 1.149 | 0.6658 | 1.072 |
| Quadruplet-TSST | 5.3 | 8.6 | 356 | 1.116 | 0.6510 | 1.057 |
| Quadruplet-STTS | 5.3 | 8.6 | 356 | 1.118 | 0.6507 | 1.057 |

| Method | Paras(M) | Flops(G) | FPS | MSE ↓ | MAE ↓ | SSIM ↑ |
|---|---|---|---|---|---|---|
| **Recurrent-based** | | | | | | |
| ConvLSTM | 15.0 | 20.7 | 815 | 0.485 | 17.7 | 0.978 |
| PredRNN | 23.7 | 42.4 | 416 | 0.464 | 16.9 | 0.977 |
| PredRNN++ | 38.4 | 63.0 | 301 | 0.448 | 16.9 | 0.971 |
| MIM | 37.9 | 64.1 | 275 | 0.429 | 16.6 | 0.971 |
| E3D-LSTM | 51.0 | 98.2 | 60 | 0.432 | 16.9 | 0.979 |
| PhyDNet | 3.1 | 5.6 | 982 | 0.362 | 15.53 | 0.983 |
| PredRNNv2 | 23.7 | 42.6 | 378 | 0.383 | 15.55 | 0.983 |
| **Recurrent-free** | | | | | | |
| SimVP | 13.8 | 3.6 | 533 | 0.414 | 16.2 | 0.982 |
| TAU | 9.6 | 2.5 | 1268 | 0.344 | 15.6 | 0.983 |
| **PredFormer** | | | | | | |
| Full Attention | 8.4 | 2.4 | 1455 | 0.316 | 14.6 | 0.985 |
| Fac-S-T | 8.4 | 2.2 | 1859 | 0.320 | 15.2 | 0.984 |
| Fac-T-S | 8.4 | 2.2 | 1839 | 0.283 | 14.4 | 0.985 |
| Binary-TS | 8.4 | 2.2 | 1773 | 0.286 | 14.6 | 0.985 |
| **Binary-ST** | 8.4 | 2.2 | 1813 | **0.277** | **14.3** | **0.986** |
| Triplet-TST | 6.3 | 1.6 | 2392 | 0.293 | 14.7 | 0.985 |
| **Triplet-STS** | 6.3 | 1.6 | 2364 | **0.277** | **14.3** | **0.986** |
| Quadruplet-TSST | 8.4 | 2.2 | 1804 | 0.284 | 14.4 | **0.986** |
| Quadruplet-STTS | 8.4 | 2.2 | 1795 | 0.293 | 14.6 | 0.985 |

**Variants of PredFormer.** In our proposed variants, several trends emerged: (1) 200-epoch experiments with patch size 4: The Fac-T-S model outperforms the full-attention model, surpassing the Fac-S-T model. The interleaved models perform significantly better than both factorized and full-attention models, with MSE values ranging from 20 to 21. Among these, the Triplet-TST model achieved the best results. (2) 200-epoch experiments with patch size 8: The interleaved models consistently outperformed both full-attention and factorized models, with a clear pattern emerging: temporal-first models performed better than spatial-first models. Notably, Quaddroplet-TSST outperformed Quaddroplet-STTS, Triplet-STS outperformed Triplet-TST, and Binary-TS slightly outperformed Binary-ST. This suggests that for the long-term 10→10 prediction task with patch size 8, temporal dependencies play a more critical role. (3) 2000-epoch experiments with patch size 4: Triplet-STS slightly outperforms Triplet-TST, achieving an MSE of 11.6. This difference may be attributed to the longer spatial sequence with a smaller patch size, where spatial dependencies become more important. (3) 2000-epoch experiments with patch size 8: Quaddroplet-TSST outperforms Triplet-TST and Binary-TS and achieves an MSE of 12.5.

## 4.2 LONG-TERM PREDICTION: WEATHERBENCH

**WeatherBench.** Climate prediction is a critical challenge in spatiotemporal predictive learning. The WeatherBench (Rasp et al., 2020) dataset provides a comprehensive global weather forecasting resource, covering various climatic factors. In our experiments, we utilize *WeatherBench-S*, a single-variable setup where each climatic factor is trained independently. We focus on temperature prediction at a $5.625°$ resolution ($32 \times 64$ grid points). The model is trained on data spanning 2010-2015, validated on data from 2016, and tested on data from 2017-2018, all with a one-hour temporal interval. We input the first 12 frames and predict the subsequent 12 frames in this setting.

**Quantitative Evaluaition.** Our quantitative on WeatherBench are shown in Tab 4. We have the following findings: (1) The first conclusion aligns with Moving MNIST, the Fac-T-S model outperforms the full attention model, which in turn outperforms the Fac-S-T model. The Fac-T-S model achieves the best overall performance with an MSE of 1.100. (2) Besides, the six interleaved models significantly outperform all other baselines by a notable margin, with MSE values ranging from 1.108 to 1.149. Notably, the Triplet-TST model achieves the second-best result 1.108. (3) The Fac-T-S model shows an 11.1% improvement over SimVP and a 5.9% improvement over TAU in terms of MSE. (4) Interestingly, the best Fac-T-S model and second-best Triplet-TST model both start with temporal blocks. Triplet-TST, which emphasizes temporal dependencies more than spatial

Table 6: Ablation study on **Gate Linear Unit** and **Position Encoding**.

| Model | Moving MNIST | | WeatherBench (T2m) | | | TaxiBJ | |
|---|---|---|---|---|---|---|---|
| | MSE ↓ | MAE ↓ | MSE ↓ | MAE ↓ | RMSE ↓ | MSE ↓ | MAE ↓ |
| PredFormer | **20.5** | **63.1** | **1.100** | **0.6489** | **1.049** | **0.277** | **14.3** |
| SwiGLU → MLP | 22.6 | 67.9 | 1.171 | 0.6707 | 1.082 | 0.306 | 15.1 |
| PE: Abs → Learnable | 22.2 | 66.7 | 1.164 | 0.6771 | 1.079 | 0.288 | 14.6 |

Table 7: Ablation study on **Dropout** and **Stochastic Depth**.

| Model | WeatherBench (T2m) | | | TaxiBJ | |
|---|---|---|---|---|---|
| | MSE ↓ | MAE ↓ | RMSE ↓ | MSE ↓ | MAE ↓ |
| Wo Reg | 1.244 | 0.7057 | 1.115 | 0.319 | 15.1 |
| + DP | 1.210 | 0.6887 | 1.100 | 0.283 | 14.5 |
| + Uni SD | 1.156 | 0.6573 | 1.075 | 0.288 | 14.6 |
| + DP + Linear SD | 1.138 | 0.6533 | 1.067 | 0.299 | 14.8 |
| + DP + Uni SD | **1.100** | **0.6489** | **1.049** | **0.277** | **14.3** |

ones, achieves comparable results with fewer parameters than Fac-T-S. This suggests that temporal dependencies are more critical for this 12→12 long-term prediction task.

**Efficiency.** Our Fac-T-S model model delivers strong performance and requires fewer parameters (reduced from 14.8M to 5.3M). Although the Fac-T-S model has comparable FLOPs (8.5G) to SimVP (8.6G), it increases the FPS from 196 to 404. Additionally, the second-best Binary-TST model excels in both efficiency and performance. These findings indicate that our model holds substantial promise for real-world weather forecasting applications.

### 4.3 SHORT-TERM PREDICTION: TAXIBJ

**TaxiBJ.** TaxiBJ (Zhang et al., 2017a) includes GPS data from taxis and meteorological data in Beijing. Each data frame is visualized as a $32 \times 32 \times 2$ heatmap, where the third dimension encapsulates the inflow and outflow of traffic within a designated area. Following previous work (Zhang et al., 2017a), we allocate the final four weeks' data for testing, utilizing the preceding data for training. Our prediction model uses four sequential observations to forecast the subsequent four frames.

**Quantitative Evaluation.** In Tab 5, we present the quantitative results on TaxiBJ. We have the following findings: (1) Among the full attention and factorized encoder models, the Fac-T-S model outperforms the full attention model, which in turn outperforms the Fac-S-T model. (2) The interleaved models outperform the full attention, Fac-S-T models, and all other baseline methods by a significant margin, with MSE values ranging from 0.277 to 0.293. Notably, Binary-ST and Triplet-STS deliver the best performance. (3) The Triplet-STS model demonstrates a 33.1% improvement over SimVP and a 19.5% improvement over TAU in terms of MSE. (4) Interestingly, both top-performing models start with spatial blocks, and Triplet-STS, which emphasizes spatial dependencies more than temporal ones, achieves comparable results with fewer parameters than Binary-ST. This suggests that spatial dependencies are more critical for this 4→4 short-term prediction task.

**Efficiency.** Our Triplet-STS model achieves superior predictive performance with fewer parameters, lower FLOPs, and higher FPS than all baselines. PredFormer reduces SimVP's parameters from 13.8M to 6.3M, FLOPs from 3.6G to 1.6G, and boosts FPS from 533 to 2364. These results underscore the model's substantial potential for real-world traffic flow prediction.

### 4.4 ABLATION STUDY AND DISCUSSION

We conduct ablation studies on our PredFormer model design and summarize the results in Tab 6. We choose the best Triplet-TST-ps4 200-epoch model on Moving MNIST, the best Triplet-STS model on TaxiBJ, and the best Fac-T-S model on WeatherBench as baselines.

**Gate Linear Unit.** Replacing SwiGLU with a standard MLP results in a notable performance degradation. On Moving MNIST, the MSE rises from 20.5 to 22.6, on TaxiBJ from 0.277 to 0.306, and on WeatherBench from 1.100 to 1.171. This consistent performance degradation highlights the critical role of the gating mechanism in modeling complex spatiotemporal dynamics.

**Position Encoding.** Additionally, the performance deteriorates when we replace the absolute positional encoding in our model with the learnable spatiotemporal encoding commonly used in ViT. On Moving MNIST, the MSE rises from 20.5 to 22.2, on TaxiBJ from 0.277 to 0.288, and on WeatherBench from 1.100 to 1.164. These ablation experiments consistently reveal similar trends across all three datasets, emphasizing the robustness of our Position Encoding designs.

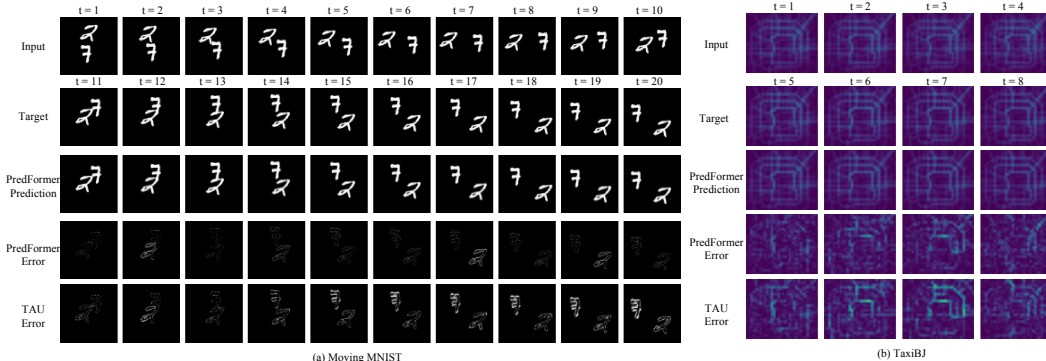

Figure 5: Visualizations on (a) Moving MNIST and (b) TaxiBJ. Error = |Prediction − Target|. We amplify the error for better comparison.

**Model Regularization.** Pure transformer architectures like ViT generally require large datasets for effective training, and overfitting can become challenging when applied to smaller datasets. In our experiments, overfitting is noticeable on WeatherBench and TaxiBJ. We experiment with different regularization techniques in Tab 7 and find that both dropout(DP) and stochastic depth (SD) individually improve performance compared to no regularization. However, the combination of the two provides the best results. Unlike conventional ViT practices, which use a linearly scaled drop path rate across different depths, a uniform drop path rate performs significantly better for our tasks. We adopt the exact regularization setting for all nine variants.

**Visualization.** Fig 5 and Appendix Fig 6 provide a visual comparison of our PredFormer model's prediction results and the associated prediction errors on three benchmark datasets. The visualizations demonstrate that our PredFormer model markedly reduces prediction errors compared to those from TAU and has more accurate predictions. We present an additional case in the Appendix Fig 7 to further demonstrate PredFormer's superior generalization ability compared to TAU.

**Discussion for PredFormer Recipe.** Despite our in-depth analysis of the spatiotemporal decomposition, the optimal model is not definite due to the different spatiotemporal dependent properties of the datasets. Within this research, long-term prediction typically emphasizes temporal dependencies, whereas short-term prediction relies more on spatial dependencies. We recommend starting with the Quadruplet-TSST model for diverse spatiotemporal prediction tasks, which consistently performs well across datasets and configurations. Use M Quadruplet-TSST layers and experiment with models having a total of 4M GTBs to identify the optimal configuration. Then, explore Triplet-TST and Triplet-TST with M layers to find spatial and temporal dependencies. Unlike SimVP framework, which adjusts hidden dimensions and block numbers separately for spatial encoder-decoder and temporal translator, PredFormer uses fixed hyperparameters for spatial and temporal GTBs, leveraging the scalability of the Transformer architecture. By simply adjusting the number of PredFormer layers, optimal results can be achieved with minimal tuning.

## 5 CONCLUSION

In this paper, we introduce PredFormer, a recurrent-free and convolution-free model designed for spatiotemporal predictive learning. Our in-depth analysis extends the understanding of spatial-temporal transformer factorization, moving beyond existing video ViT frameworks. Through rigorous experiments, PredFormer shows unparalleled performance and efficiency, surpassing previous models by a large margin. Our results elucidate several critical insights: (1) Interleaved spatiotemporal transformer architectures establish new benchmarks, excelling across multiple datasets. (2) Factorized temporal-first encoders significantly outperform both full spatial-temporal attention encoders and Factorized spatial-first configurations. (3) Implementing dropout and uniform stochastic depth concurrently leads to superior performance enhancements on overfitting datasets. (4) Absolute position encoding consistently outperforms learnable alternatives across all benchmarks. We believe PredFormer will not only establish a robust baseline for real-world applications but also pave the way for future innovations in pure transformer-based spatiotemporal predictive models.

**Reproducibility Statement**   We provide detailed instructions for implementing our method and reproducing the experiments in Sec 4 and Appendix Sec A.2. Our experiments use open-source datasets, and we will release the code and trained models to the public upon acceptance.

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

# A APPENDIX

## A.1 PROBLEM DEFINITION

Spatiotemporal predictive learning is to learn spatial and temporal patterns by predicting future frames based on past observations. Given a sequence of frames $\mathcal{X}^{t:T} = \{\boldsymbol{x}^i\}_{t-T+1}^{t}$, which encapsulates the last $T$ frames leading up to time $t$, the goal is to forecast the following $T'$ frames $\mathcal{Y}^{t+1:T'} = \{\boldsymbol{x}^i\}_{t+1}^{t+1+T'}$ starting from time $t+1$. The input and the output sequence are represented as tensors $\mathcal{X}^{t:T} \in \mathbb{R}^{T \times C \times H \times W}$ and $\mathcal{Y}^{t+1:T'} \in \mathbb{R}^{T' \times C \times H \times W}$, where $C$, $H$, and $W$ denote channel, height, and width of frames, respectively. The $T$ and $T'$ are the input and output frame numbers. For brevity, we use $\mathcal{X}$ and $\mathcal{Y}$ to denote $\mathcal{X}^{t:T}$ and $\mathcal{Y}^{t+1:T'}$ in the following sections.

Generally, we adopt a deep model equipped with learnable parameters $\mathcal{F}_\Theta$ for future frame prediction. The optimal set of parameters $\Theta^*$ is obtained by solving the optimization problem:

$$\Theta^* = \arg\min_\Theta \mathcal{L}(\mathcal{F}_\Theta(\mathcal{X}), \mathcal{Y}) \tag{4}$$

where $\mathcal{L}$ is the loss function measuring the difference between the prediction and the ground truth.

## A.2 EXPERIMENT SETTING

For the 200-epoch Moving MNIST experiment with a patch size of 4, we use a batch size of 2 for the full attention model and a batch size of 8 for other variants due to memory constraints. For the 2000-epoch experiment with the same patch size, we increase the batch size to 16, utilizing a single 48GB A6000 GPU. In experiments with a patch size of 8, we maintain a batch size of 16 on a 24GB GPU across all runs. For Moving MNIST, we use 24 GTB blocks for all PredFormer variants, which means 6 Quadruplet-TSST layers, 8 Triplet-TST layers, and 12 Binary-TS layers, respectively.

For the TaxiBJ and WeatherBench datasets, we use 6 GTB blocks for the Triplet variants and 8 GTB blocks for the other variants.

Table 8: Hyperparameter Setting.

| | Moving MNIST | TaxiBJ | WeatherBench |
|---|---|---|---|
| **Training Hyperparameter** | | | |
| Batch Size | {8,16} | 16 | 16 |
| Learning Rate | 1e-3 | 1e-3 | 5e-4 |
| Learning Scheduler | Onecycle | Onecycle | Cosine |
| Optimizer | Adamw | Adamw | Adamw |
| Weight Decay | 1e-2 | 1e-2 | 1e-2 |
| Training Epochs | {200,2000} | 200 | 50 |
| **Model Hyperparameter** | | | |
| Patch Size | {4,8} | 4 | 4 |
| GTB Blocks | 24 | {6,8} | {6,8} |
| GTB Dim | 256 | 256 | 256 |
| GTB Heads | 8 | 8 | 8 |
| SwiGLU Hidden Dim | 1024 | 1024 | 512 |
| Attention Dropout | 0.0 | 0.1 | 0.1 |
| SwiGLU Dropout | 0.0 | 0.1 | 0.1 |
| Drop Path Rate | 0.0 | 0.1 | 0.25 |

## A.3 MORE VISUALIZATIONS

Figures 6 shows the visualization on WeatherBench. As the number of frames increases, TAU's error increases more significantly compared to ours. This demonstrates the strength of our PredFormer model for long-term forecasting

Figures 7(a) and (b) depict the inflow and outflow at the same time step. In this case, the fourth frame shows significantly less traffic flow than the previous frames. Constrained by the inductive

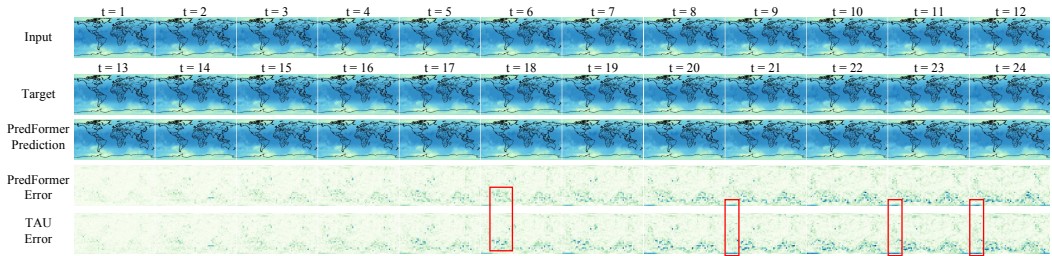

Figure 6: Visualizations on WeatherBench for global temperature forecasting.

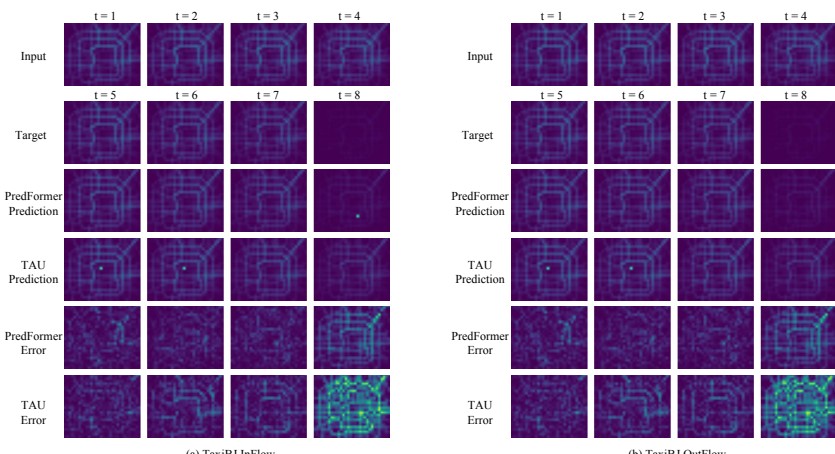

Figure 7: Visualizations on TaxiBJ Inflow and OutFlow. We amplify the error for better comparison.

bias of CNNs, TAU continues to predict high traffic levels. In contrast, our PredFormer demonstrates superior generalization by accurately capturing this abrupt change. This capability highlights PredFormer's potential to handle extreme cases, which could be particularly valuable in applications like traffic flow prediction and weather forecasting.

## A.4    MORE EXPERIMENTS

We provide additional experimental results to further validate the effectiveness and efficiency of PredFormer compared to existing methods. Tab 9 showcases the performance of PredFormer against transformer-based spatiotemporal prediction models, including SwinLSTM and OpenSTL, on the Moving MNIST dataset, demonstrating its faster training time and superior accuracy. Tab 11 highlights the comparison of PredFormer with SwinLSTM and OpenSTL on the TaxiBJ dataset, illustrating PredFormer's significantly higher FPS and lower MSE. Tab 12 compares PredFormer with various existing methods on the Human3.6M dataset, showcasing its competitive accuracy with superior efficiency in terms of FLOPs and FPS. Tab 13 illustrates the comparison between Pred-Former and EarthFormer on the Moving MNIST dataset, highlighting PredFormer's efficiency with lower FLOPs and better performance. Tab 14 presents an ablation study of PredFormer by varying the number of TSST layers, showing that even with fewer layers, PredFormer achieves better results compared to competing methods like SimVP and TAU. Finally, Tab 15 and Tab 16 compare PredFormer with SwinLSTM and VMRNN on Moving MNIST and TaxiBJ datasets, respectively, emphasizing its faster training and inference speeds, as well as its ability to deliver lower MSE and higher SSIM with comparable or fewer parameters and FLOPs.

Table 9: Comparisons of PredFormer and SwinLSTM on the Moving MNIST dataset with 2000 training epochs.

| Method | Paras (M) | Flops (G) | Training Epoch Time | MSE | SSIM |
|---|---|---|---|---|---|
| SwinLSTM | – | – | 9min | 17.7 | 0.962 |
| PredFormer | 25.3 | 16.5 | 3.5min | **12.5** | **0.973** |

Table 10: Comparisons of PredFormer and OpenSTL on the Moving MNIST dataset with 200 training epochs.

| Method | Paras (M) | Flops (G) | MSE | MAE | SSIM |
|---|---|---|---|---|---|
| OpenSTL+ViT | 46.1 | 16.9 | 35.2 | 95.9 | 0.914 |
| OpenSTL+Swin Transformer | 46.1 | **16.4** | 29.7 | 84.1 | 0.933 |
| PredFormer | **25.3** | 16.5 | **26.0** | **77.2** | **0.941** |

Table 11: Comparisons of PredFormer, SwinLSTM, and OpenSTL on the TaxiBJ dataset.

| Method | Paras (M) | Flops (G) | FPS | MSE | SSIM |
|---|---|---|---|---|---|
| SwinLSTM | **2.9** | **1.3** | 1425 | 0.303 | 0.984 |
| OpenSTL+ViT | 9.7 | 2.8 | 1301 | 0.317 | 0.984 |
| OpenSTL+Swin Transformer | 9.7 | 2.6 | 1506 | 0.313 | 0.984 |
| PredFormer | 6.3 | 1.6 | **2364** | **0.277** | **0.986** |

Table 12: Comparisons of PredFormer and OpenSTL on the Human3.6M dataset.

| Method | Paras (M) | Flops (G) | FPS | MSE | MAE |
|---|---|---|---|---|---|
| OpenSTL+ViT | 28.3 | 239.0 | 17 | 136.3 | 1603.5 |
| OpenSTL+Swin Transformer | 38.8 | 188.0 | 28 | 133.2 | 1509.7 |
| PredFormer | **12.7** | **65.2** | **78** | **114.7** | **1403.6** |

Table 13: Comparison of PredFormer and EarthFormer on the Moving MNIST dataset.

| Method | Paras (M) | Flops (G) | MSE | MAE | SSIM |
|---|---|---|---|---|---|
| EarthFormer | **6.6** | 33.7 | 46.9 | 101.5 | 0.883 |
| PredFormer 2TSST Layer | 8.5 | **5.5** | 20.1 | 65.3 | 0.955 |
| PredFormer 6TSST Layer | 25.3 | 16.5 | **12.5** | **44.6** | **0.973** |

Table 14: Ablation study of PredFormer layer number on the Moving MNIST dataset.

| Method | Paras (M) | Flops (G) | FPS | MSE | MAE | SSIM |
|---|---|---|---|---|---|---|
| SimVP | 58.0 | 19.4 | 209 | 23.8 | 68.9 | 0.948 |
| TAU | 44.7 | 16.0 | 283 | 19.8 | 60.3 | 0.957 |
| PredFormer 3TSST Layer | **12.7** | **8.3** | **291** | 16.2 | 55.1 | 0.965 |
| PredFormer 6TSST Layer | 25.3 | 16.5 | 152 | **12.5** | **44.6** | **0.973** |

Table 15: Comparisons of PredFormer, SwinLSTM, and VMRNN on the Moving MNIST dataset.

| Method | Paras (M) | Flops (G) | Epoch Time | MSE | SSIM |
|---|---|---|---|---|---|
| SwinLSTM | – | – | 9min | 17.7 | 0.962 |
| VMRNN | – | – | 18min | 16.5 | 0.965 |
| PredFormer 3TSST Layer | 12.7 | 8.3 | **1.5min** | 16.2 | 0.965 |
| PredFormer 6TSST Layer | 25.3 | 16.5 | 3.5min | **12.5** | **0.973** |

Table 16: Comparison of PredFormer, SwinLSTM, and VMRNN on the TaxiBJ dataset.

| Method | Paras (M) | Flops (G) | Epoch Time | FPS | MSE | MAE | SSIM |
|---|---|---|---|---|---|---|---|
| SwinLSTM | 2.9 | 1.3 | – | 1425 | 0.303 | 15.0 | 0.9843 |
| VMRNN | **2.6** | **0.9** | 5min | 526 | 0.289 | 14.7 | 0.9858 |
| PredFormer | 6.3 | 1.6 | **1min** | **2354** | **0.277** | **14.3** | **0.9864** |

