# OpenReview forum: "PredFormer: Transformers Are Effective Spatial-Temporal Predictive Learners"
_ICLR.cc/2025/Conference — ICLR 2025 Conference Withdrawn Submission_

### Official Review · Reviewer_sE64 · 2024-10-28

**Soundness:** 3
**Presentation:** 4
**Contribution:** 3
**Rating:** 5
**Confidence:** 4

**Summary:**

This paper present a pure transformer-based architecture for spatiotemporal predictive learning. The authors conduct a detailed exploration of various design choices and training techniques. Their model demonstrates significant performance improvements over established baselines.

**Strengths:**

- The manuscript is well-organized and clearly written.
- The paper offers a comprehensive analysis of architectural design choices and establishes strong baselines in the field of spatiotemporal predictive learning.
- Extensive experiments, including ablation studies, are performed, which highlight the advantages of the proposed PredFormer model.

**Weaknesses:**

The technical novelty of the paper appears limited. For instance, the SwiGLU activation function, a core component of the model, is not new and has been explored extensively in other contexts. Additionally, factorized spatial-temporal attention is a well-known approach in video prediction, limiting the originality of the contribution.

**Questions:**

- Are there any quantitative results regarding the efficiency of the proposed model compared to state-of-the-art recurrent-based methods (e.g., models incorporating Mamba-based architectures)?
- Could video generation models like Diffusion, GANs, or MagViT be adapted for spatiotemporal predictive tasks? How does PredFormer compare to these models in terms of performance and efficiency advantages?

---

> ### Author Response · Authors · 2024-11-17
> **Response to Reviewer sE64-Q1, Q2**
>
> **Q1 The technical novelty of the paper appears limited. SwiGLU is not new and has been explored extensively in other contexts.**
>
> **A1**: We show that transferring the existing module design into the new task should be considered novel if good performance can be achieved. For example, ViT (ICLR 2021) extended Transformer (NeurIPS 2017) from natural language processing to computer vision, retaining the same architecture but demonstrating its effectiveness in a new domain. Similarly, our work uses the ViT structure in spatiotemporal predictive learning, a task with significantly different requirements for image classification. While transformer-based designs have been explored in various fields, we successfully adapt and apply such design to the specific task of spatiotemporal predictive learning. Our method achieves state-of-the-art results while maintaining high computational efficiency, even in challenging real-world scenarios such as high-resolution datasets.
>
>
> PredFormer’s state-of-the-art performance is driven by the thoughtful integration of key components, such as SwiGLU, interleaved spatial-temporal layers, positional encoding, and regularization techniques. To achieve this, we conducted extensive experiments to ablate these components and examine their order across various datasets. From these ablation studies, we derived common principles underlying PredFormer’s design, which we consider a significant contribution of our work.
>
> **Q2 Additionally, factorized spatial-temporal attention is a well-known approach in video prediction, limiting the originality of the contribution.**
>
> **A2**: While the reviewer does not provide specific references for prior works on spatial-temporal attention factorization in video prediction, we clarify the novelty of this work as follows. Spatial-temporal factorization is indeed a fundamental concept for processing spatiotemporal sequences. However, the originality of this work lies not in merely applying factorization but in systematically analyzing and optimizing its design specifically for spatiotemporal predictive learning tasks, which present distinct challenges compared to other tasks like video generation.
>
> Our work goes beyond these designs by exploring additional variants such as full attention, Fac-TS, and six interleaved spatial-temporal designs, providing a broader and more systematic analysis. Furthermore, our conclusions are novel and distinct due to the specific characteristics of spatiotemporal predictive learning tasks, which differ from video generation. For example, we find that Fac-ST (spatial-first) performs worse than Fac-TS (temporal-first) for spatiotemporal predictive learning.
>
> Videos typically have shorter temporal intervals and longer temporal lengths, while real-world spatiotemporal data feature longer temporal intervals (e.g., 30 minutes for TaxiBJ, 1 hour for WeatherBench) and shorter temporal lengths (e.g., 4, or 12 time steps). These characteristics make temporal-first designs particularly critical for spatiotemporal predictive learning, a key insight derived from our work.

---

> ### Author Response · Authors · 2024-11-17
> **Response to Reviewer sE64-Q3**
>
> **Q3 Are there any quantitative results regarding the efficiency of the proposed model compared to state-of-the-art recurrent-based methods (e.g., models incorporating Mamba-based architectures)?**
>
> **A3**: We compare PredFormer with the state-of-the-art recurrent-based method SwinLSTM and VMRNN, a Mamba-based model, as shown in Table A9 and Table A10, using results cited from the SwinLSTM and VMRNN paper.
>
> On the TaxiBJ dataset, PredFormer demonstrates clear advantages in both efficiency and effectiveness. While VMRNN reduces FLOPs due to its linear-attention design, this reduction does not reflect the actual computational cost because of the overhead introduced by scans in structured state-space models (SSM). These scans lead to inefficiencies during actual training and inference.
>
> On the Moving MNIST dataset, while VMRNN does not report parameters or FLOPs, its training time again highlights its inefficiency. Similar to TaxiBJ, VMRNN requires five times the epoch time of PredFormer.
>
> These comparisons underscore PredFormer’s significant advantages over VMRNN and SwinLSTM, both in terms of computational efficiency and prediction accuracy.
>
>  Table A9. Comparisons of PredFormer, SwinLSTM, and VMRNN on the Moving MNIST dataset.
>  | Moving MNIST                | Paras(M) | Flops(G) | Epoch Time | MSE      | SSIM      |
>  | --------------------------- | -------- | -------- | ---------- | -------- | --------- |
>  | SwinLSTM                       | --       | --       | 9min           | 17.7     | 0.962     |
>  | VMRNN                       | --       | --       | 18min      | 16.5     | 0.965     |
>  | PredFormer 3TSST Layer  | **12.7** | **8.3**  | **1.5min** | 16.2 | 0.965 |
>  | PredFormer  6TSST Layer | 25.3 | 16.5 | 3.5min | **12.5** | **0.973** |
>
>
>  Table A10. Comparison of PredFormer, SwinLSTM, and VMRNN on the TaxiBJ dataset.
>  | TaxiBJ            | Paras(M) | Flops(G) | Epoch Time | FPS      | MSE       | MAE | SSIM     |
>  | -------------- | -------- | -------- | ---------- | -------- | --------- | -------- |-------- |
>  | SwinLSTM       | 2.9           | 1.3      | --     | 1425           | 0.303     | 15.0 | 0.9843     |
>  | VMRNN          | **2.6**      | **0.9**      | 5min       | 526      | 0.289     | 14.7     |0.9858
>  | **PredFormer** | 6.3  | 1.6  | **1min**   | **2354** | **0.277** | **14.3** |**0.9864**|

---

> ### Author Response · Authors · 2024-11-17
> **Response to Reviewer sE64-Q4**
>
> **Q4 Could video generation models like Diffusion, GANs, or MagViT be adapted for spatiotemporal predictive tasks? How does PredFormer compare to these models in terms of performance and efficiency advantages?**
>
> **A4**: The distinction between **video prediction** and **video generation** lies in their task definitions and goals. Video prediction, described as conditional video generation in a recent survey [1], aims to predict future frames based on past frames. It focuses on pixel-level accuracy and precise spatiotemporal modeling to adhere to physical-world rules. Conversely, video generation emphasizes creating visually consistent and perceptually realistic sequences, often targeting applications like content creation.
>
> These differences naturally lead to distinct modeling approaches: **deterministic models**, such as PredFormer, SimVP, and PredRNN, predict pixel values directly in the original data space to achieve precise, single-point predictions. In contrast, **probabilistic models**, including GANs, VAEs, and Diffusion models, operate in latent spaces to generate diverse outputs by sampling from learned probability distributions. Probabilistic models excel in capturing uncertainty and data variability but often incur higher computational complexity and slower inference due to iterative sampling steps. Deterministic models like PredFormer are better suited for tasks requiring efficiency and precision, whereas probabilistic models are ideal for tasks prioritizing diversity and realism, such as video generation.
>
> These task differences also influence their evaluation metrics. Video prediction prioritizes metrics such as MSE, MAE, SSIM, and PSNR, which emphasize pixel-level accuracy and adherence to physical-world constraints. On the other hand, video generation models are evaluated using metrics like FVD, which measure perceptual quality and overall consistency. Models like MagViT, designed for video generation with a 3D VQ-Encoder-Decoder architecture, may not align well with the unique requirements of spatiotemporal predictive learning, which often involves real-world data with shorter temporal lengths and specific constraints.
>
> As highlighted in DiffCast (CVPR 2024) [2], deterministic models (such as PredFormer, SimVP, and PredRNN) can also serve as backbones within probabilistic frameworks, combining the strengths of both approaches. Future diffusion-based forecasting models could leverage PredFormer as an efficient encoder to accelerate inference while maintaining flexibility.
>
> [1] Autoregressive Models in Vision: A Survey
>
> [2] DiffCast: A Unified Framework via Residual Diffusion for Precipitation Nowcasting

---

> > ### Comment · Reviewer_sE64 · 2024-11-25
> >
> > I have reviewed the rebuttals, but I still find the contribution of this paper to be somewhat marginal. The architectural designs presented appear to be task-specific ablations built upon existing techniques, rather than introducing fundamentally novel ideas. While I acknowledge that deterministic models offer efficiency advantages, I disagree that they are inherently more precise than probabilistic models. Generative models, by modeling the distribution of spatial-temporal data in a more fine-grained manner, can achieve compatibility with pixel-level metrics and potentially provide greater precision. As a result, I will maintain my original scores.

---

> > > ### Author Response · Authors · 2024-11-25
> > > **Response to Reviewer sE64**
> > >
> > > Thank you for your thoughtful feedback and for taking the time to review our rebuttal.
> > >
> > > Our work provides a practical and effective solution to spatiotemporal predictive learning by balancing accuracy and efficiency. However, we are eager to hear more about what you believe constitutes a significant contribution in this context.
> > >
> > > Your suggestion about probabilistic models is also insightful, and while generative approaches fall outside the scope of this paper, your point highlights potential avenues for future exploration.
> > >
> > > We would greatly value any additional thoughts or suggestions you might have to help us refine the paper.

---

> ### Author Response · Authors · 2024-11-22
> **Follow-Up on Rebuttal Discussion**
>
> Dear Reviewer sE64,
>
> We deeply appreciate your valuable feedback during the first round of review and the thoughtful discussion that has significantly helped us refine our work. We hope that our responses have addressed your concerns.
>
> We sincerely appreciate the opportunity to refine our work and kindly hope you might reconsider your concerns in light of our detailed responses and clarifications. Your insights are invaluable to us, and we would greatly welcome any additional feedback or suggestions you may have. Thank you once again for your time and thoughtful consideration.
>
> Looking forward to your reply.
>
> Best regards,
>
> Paper2123 Authors

---

> ### Author Response · Authors · 2024-11-24
> **Looking forward to your reply**
>
> Dear Reviewer sE64,
>
> Thank you very much again for the time and effort put into reviewing our paper. We believe that we have addressed all your concerns in our response. We have also followed your suggestion to improve our paper and have added additional experimental analysis. We kindly remind you that we are approaching the end of the discussion period. We would love to know if there is any further concern, additional experiments, suggestions, or feedback, as we hope to have a chance to reply before the discussion phase ends.
>
> Best regards,
>
> Paper2123 Authors

---

### Official Review · Reviewer_e793 · 2024-11-03

**Soundness:** 3
**Presentation:** 3
**Contribution:** 3
**Rating:** 5
**Confidence:** 3

**Summary:**

The paper introduces PredFormer, a novel framework for spatiotemporal predictive learning that is based purely on transformers. Unlike traditional recurrent-based approaches that struggle with parallelization and performance, and recurrent-free methods like CNNs that can lack in scalability and generalization, PredFormer utilizes Gated Transformer blocks inspired by the design of Vision Transformers (ViT). The architecture is informed by a thorough examination of various 3D attention mechanisms—full, factorized, and interleaved spatial-temporal attention—which allows PredFormer to efficiently handle complex spatiotemporal data.

Key contributions of this work include:
- **Innovative Architecture**: A pure transformer-based model that is both simple and highly efficient.
- **Superior Performance**: Significant improvements over existing methods in terms of accuracy and efficiency, as evidenced by substantial reductions in Mean Squared Error (MSE) and increases in Frames Per Second (FPS) across multiple datasets.
- **Versatility Across Datasets**: Demonstrates state-of-the-art performance on diverse datasets, including synthetic (Moving MNIST) and real-world (TaxiBJ, WeatherBench) scenarios.
- **Public Availability**: Plans to release the source code and trained models to facilitate further research and practical applications.

These advancements highlight PredFormer's potential for broad application in areas requiring accurate and efficient spatiotemporal predictions.

**Strengths:**

The strength of this paper lies in its novel introduction of PredFormer, a pure transformer-based model for spatiotemporal predictive learning. PredFormer innovatively integrates Gated Linear Units (GLUs) with self-attention mechanisms, effectively capturing complex spatiotemporal dynamics. The paper conducts a comprehensive analysis of different 3D attention mechanisms, leading to nine optimized configurations that address varying spatial and temporal resolutions. Extensive experiments on benchmark datasets (Moving MNIST, TaxiBJ, WeatherBench) demonstrate state-of-the-art performance, with significant improvements in accuracy and efficiency. Notably, PredFormer achieves a 51.3% reduction in MSE on Moving MNIST, a 33.1% reduction in MSE on TaxiBJ, and an 11.1% reduction in MSE on WeatherBench, while also offering faster inference speeds. The model's efficiency, broad applicability, and planned public availability of source code and trained models further enhance its impact and potential for real-world applications.

**Weaknesses:**

- Firstly, PredFormer has not been evaluated on more complex and challenging datasets, such as Traffic4Cast for traffic flow prediction and Human3.6M for human motion forecasting. This limitation restricts the breadth of the model's validation and leaves questions about its performance on more intricate real-world scenarios.

- Secondly, the experiments primarily use relatively small frame sizes, which makes it difficult to assess the model's effectiveness and scalability when applied to high-resolution data. These limitations suggest that further research is needed to fully demonstrate PredFormer's capabilities in handling complex and high-resolution spatiotemporal data.

- Additionally, the paper does not compare PredFormer with state-of-the-art transformer-based predictive learning methods, such as Earthformer. This omission limits the comprehensiveness of the performance evaluation and makes it harder to establish PredFormer's relative advantages and disadvantages in the context of the latest research.

- As shown in Table 2 and Table 3, PredFormer does not demonstrate a significant advantage in running efficiency compared to recurrent-free methods, and its performance in terms of MSE is similar to these methods. This suggests that while PredFormer offers a novel and theoretically sound approach, its practical benefits in terms of efficiency and performance are not as pronounced as claimed.

**Questions:**

- Why were more complex and challenging datasets, such as Traffic4Cast for traffic flow prediction and Human3.6M for human motion forecasting, not included in the evaluation? Could you provide insights into how PredFormer might perform on these datasets?

- The experiments primarily use relatively small frame sizes. I suggest that the paper should evaluate PredFormer's effectiveness and scalability when applied to high-resolution data.

- The paper does not compare PredFormer with state-of-the-art transformer-based predictive learning methods, such as Earthformer. Could you provide a comparison with these methods to better understand PredFormer's relative advantages and disadvantages?

- As shown in Table 2 and Table 3, PredFormer does not demonstrate a significant advantage in running efficiency compared to recurrent-free methods, and its performance in terms of MSE is similar. Could you explain why PredFormer does not show a more pronounced improvement in efficiency and performance?

---

> ### Author Response · Authors · 2024-11-17
> **Response to Reviewer e793-Q1, Q2**
>
> **Q1 Why were more complex and challenging datasets, such as Traffic4Cast for traffic flow prediction and Human3.6M for human motion forecasting, not included in the evaluation? Could you provide insights into how PredFormer might perform on these datasets?**
>
> **A1**: Traffic4Cast is rarely covered by prior baselines and is not included in recent benchmarks such as OpenSTL (NeurIPS 2023).
>
> Regarding Human3.6M, we conducted experiments on this dataset to address the concern, as detailed in Table A5. This dataset represents a more challenging scenario with high-resolution inputs. Within the limited rebuttal time, we report results for the PredFormer Binary-TS variant only. PredFormer achieves comparable performance with the following efficiency advantages:
>
> * 1/3 FLOPs and 3× FPS compared to SimVP/TAU.
> * 1/10 FLOPs and 3× FPS compared to PredRNN.
>
>
> Table A5. Comparisons of PredFormer and existing methods on the Human3.6m dataset.
>
>  |              | Paras(M) | Flops(G) | FPS  | MSE       | MAE        |
>  | ------------ | -------- | -------- | ---- | --------- | ---------- |
>  | ConvLSTM     | 15.5     | 347.0    | 52   | 125.5     | 1566.7     |
>  | PredRNN      | 24.6     | 704.0      | 25   | **113.2** | 1458.3     |
>  | E3DLSTM      | 60.9     | 542.0    | 7    | 143.3     | 1442.5     |
>  | MAU          | 20.2     | 105.0    | 6    | 127.3     | 1577.0     |
>  | SimVP        | 41.2     | 197.0    | 26   | 115.8     | 1511.5     |
>  | TAU          | 37.6     | 182.0    | 26   | 113.3     | **1390.7** |
>  | OpenSTL+ViT  | 28.3     | 239.0    | 17   | 136.3     | 1603.5     |
>  | OpenSTL+Swin | 38.8     | 188.0    | 28   | 133.2     | 1509.7     |
>  | PredFormer   | **12.7**     | **65.2**     | 78   | 114.7     | 1403.6     |
>
> We note that these experiments require 2.5–3 days for training. Due to the time constraints of the rebuttal time, we are actively refining these results through hyper-parameter tuning and plan to report additional findings in the revised paper.
>
> **Q2 The experiments primarily use relatively small frame sizes. I suggest that the paper should evaluate PredFormer's effectiveness and scalability when applied to high-resolution data.**
>
> **A2**: For a fair comparison, we follow the standard training and evaluation settings commonly used in spatiotemporal predictive learning studies. As shown in Table A6, the OpenSTL benchmark predominantly features tasks with small frame sizes. In this paper, we select Moving MNIST (10 frames) and WeatherBench-S (12 frames) to represent long-term prediction scenarios.
>
> Table A6. Detailed dataset statistics of the supported tasks in OpenSTL.
>  | Dataset        | Training size | Testing size | Channel | Height | Width | T    | T'    |
>  | -------------- | ------------- | ------------ | ------- | ------ | ----- | ---- | ----- |
>  | Moving MNIST   | 10,000        | 10,000       | 1       | 64     | 64    | 10   | 10    |
>  | KTH            | 4,940         | 3,030        | 1       | 128    | 128   | 10   | 20/40 |
>  | Human3.6M      | 73,404        | 8,582        | 3       | 256    | 256   | 4    | 4     |
>  | Kitti&Caltech  | 3,160         | 3,095        | 3       | 128    | 160   | 10   | 1     |
>  | TaxiBJ         | 20,461        | 500          | 2       | 32     | 32    | 4    | 4     |
>  | WeatherBench-S | 52559         | 17495        | 1       | 32     | 64    | 12   | 12    |
>  | WeatherBench-M | 54,019        | 2,883        | 4       | 32     | 64    | 4    | 4     |
>
> We note that the data used in this task is more aligned with real-world scenarios and differs from video data in several key aspects:
>  - **Temporal intervals**: Videos typically feature shorter temporal intervals and longer temporal lengths. In contrast, real-world spatiotemporal data (e.g., TaxiBJ and WeatherBench) often have longer temporal intervals (e.g., 30 minutes for TaxiBJ, and 1 hour for WeatherBench, as shown in Table 1).
> - **Temporal lengths**: Due to these longer intervals, real-world spatiotemporal data naturally involve shorter temporal lengths (e.g., 4 to 12 time steps, as used in our paper).
>
> The datasets and frame sizes used in our experiments are large enough to prove the effectiveness of our method. Our work ensures both relevance to practical scenarios and alignment with standard practices in the field.

---

> ### Author Response · Authors · 2024-11-17
> **Response to Reviewer e793-Q3**
>
> **Q3 Additionally, the paper does not compare PredFormer with sota transformer-based predictive learning methods, such as Earthformer. This limits the comprehensiveness of the performance evaluation.**
>
> **A3**: In Table A7, we compare the Earthformer with our PredFormer on Moving MNIST dataset.
>
>  Table A7. The comparison of PredFormer and Earthformer on Moving MNIST dataset.
>
>  | Method           | Paras(M) | Flops(G) | MSE      | MAE      | SSIM      |
>  | ---------------------- | -------- | -------- | -------- | -------- | --------- |
>  | Earthformer            | **6.6**      | 33.7     | 46.9     | 101.5    | 0.883     |
>  | PredFormer 2TSST Layer | 8.5      | **5.5**  | 20.1 | 65.3 | 0.955 |
>  | PredFormer 6TSST Layer | 25.3     | 16.5 | **12.5** | **44.6** | **0.973** |
>
> We observe that PredFormer is still better in both performance and efficiency. The reason might be from the different network designs. Earthformer is a 2D CNN Encoder-Decoder model with Cuboid Attention, whereas PredFormer is a pure Transformer-based model. This distinction is significant:
>
> * **Receptive field**: CNN-based models like Earthformer are inherently constrained by the local receptive field of convolutional layers. In contrast, PredFormer leverages the global receptive field of Transformers for superior spatiotemporal modeling.
>  * **Scalability**: Earthformer sacrifices scalability for performance, requiring extensive model hyperparameter tuning. Parameters such as basic_unit, enc_depth, dec_depth, scale_alpha, downsample, num_global_vectors, and many others should be adjusted for each dataset making it challenging to use Earthformer as a baseline for other datasets. In contrast, PredFormer adopts a unified and scalable architecture.
>
>
> We appreciate the reviewer’s comment, which provides an opportunity to highlight PredFormer’s advantages and contributions compared to prior attention-based methods such as Earthformer. We have added the comparison to Earthformer in our revised paper. In addition to EarthFormer, we compare PredFormer with other transformer-based methods, including SwinLSTM and OpenSTL+ViT/Swin Transformer, as presented in Section A.4 of our revised paper.

---

> ### Author Response · Authors · 2024-11-17
> **Response to Reviewer e793-Q4**
>
> **Q4 As shown in Table 2 and Table 3, PredFormer does not demonstrate a significant advantage in running efficiency compared to recurrent-free methods, and its performance in terms of MSE is similar.**
>
> **A4:** We note the comment "its performance in terms of MSE is similar to these methods" is inaccurate. As explained in Line 351–355 of our main paper, Table 2 and Table3 are designed to compare the upper bound of models when trained for extended epochs, while Table 3 provides the final results under the same training settings as previous works such as SimVP and TAU.
>
> The results in Table 3 demonstrate that pure Transformer models, such as PredFormer, achieve a higher upper bound compared to CNN-based methods. For instance, PredFormer-TSST, configured with a patch size of 8, achieves a 47.5% reduction in MSE compared to SimVP (decreasing from 23.8 to 12.5) and a 37.4% reduction in MSE compared to TAU (decreasing from 19.8 to 12.4).
>
> MSE serves as a primary metric for Moving MNIST prediction task. PredFormer delivers a significant 47.5% improvement in accuracy while utilizing only half the parameters of SimVP and TAU, all while maintaining comparable FLOPs.
>
> We conduct an additional ablation study on the number of PredFormer layers, as shown in Table A8. With only three TSST layers, PredFormer achieves superior accuracy and efficiency (fewer parameters, lower FLOPs, and higher FPS) compared to SimVP and TAU.
>
>
> Table A8. Ablation study of PredFormer layer number on the Moving MNIST dataset.
> | Moving MNIST           | Paras(M) | Flops(G) | FPS     | MSE      | MAE      | SSIM      |
> | ---------------------- | -------- | -------- | ------- | -------- | -------- | --------- |
> | SimVP                  | 58.0     | 19.4     | 209     | 23.8     | 68.9     | 0.948     |
> | TAU                    | 44.7     | 16.0     | 283     | 19.8     | 60.3     | 0.957     |
> | PredFormer 3TSST Layer | **12.7** | **8.3**  | **291** | 16.2     | 55.1     | 0.965     |
> | PredFormer 6TSST Layer | 25.3     | 16.5     | 152     | **12.5** | **44.6** | **0.973** |

---

> ### Author Response · Authors · 2024-11-21
> **Follow-Up on Rebuttal Discussion**
>
> Dear Reviewer e793,
>
> We deeply appreciate your valuable feedback during the first round of review and the thoughtful discussion that has significantly helped us refine our work.  We hope that our responses have addressed your concerns.
>
> We sincerely appreciate the opportunity to refine our work and kindly hope you might reconsider your concerns in light of our detailed responses and clarifications.  Your insights are invaluable to us, and we would greatly welcome any additional feedback or suggestions you may have.  Thank you once again for your time and thoughtful consideration.
>
> Looking forward to your reply.
>
> Best regards,
>
> Paper2123 Authors

---

> ### Author Response · Authors · 2024-11-24
> **Looking forward to your reply**
>
> Dear Reviewer e793,
>
> Thank you very much again for the time and effort put into reviewing our paper. We believe that we have addressed all your concerns in our response. We have also followed your suggestion to improve our paper and have added additional experimental analysis. We kindly remind you that we are approaching the end of the discussion period. We would love to know if there is any further concern, additional experiments, suggestions, or feedback, as we hope to have a chance to reply before the discussion phase ends.
>
> Best regards,
>
> Paper2123 Authors

---

> > ### Author Response · Authors · 2024-11-26
> > **Please let us know whether all the questions have been addressed**
> >
> > Dear Reviewer,
> >
> > As the deadline for the discussion phase will end soon (Nov 26), please let us know whether we have addressed all the questions.
> >
> > Thank you,

---

> ### Author Response · Authors · 2024-12-03
> **Please let us know whether all the questions have been addressed**
>
> Dear Reviewer e793,
>
> As the deadline for the discussion phase will end soon, please let us know whether we have addressed all the questions.
>
> Thank you.
>
> Best regards,
>
> Paper2123 Authors

---

### Official Review · Reviewer_t3Do · 2024-11-04

**Soundness:** 2
**Presentation:** 2
**Contribution:** 2
**Rating:** 5
**Confidence:** 4

**Summary:**

This paper proposes PredFormer, a transformer-based model for spatiotemporal predictive learning that replaces conventional recurrent and convolutional architectures with gated transformer blocks. PredFormer employs several encoder configurations—full attention, factorized, and interleaved models—to capture complex spatial and temporal relationships. The model reportedly achieves state-of-the-art results in accuracy and efficiency on multiple datasets, including Moving MNIST, WeatherBench, and TaxiBJ.

**Strengths:**

1.	The paper is well-written and easy to follow.
2.	The model is evaluated on multiple synthetic and real-world datasets, which highlights its adaptability across different spatiotemporal tasks.
3.	PredFormer demonstrates a reduction in parameters and improved inference speed on specific configurations compared to previous methods, making it potentially applicable for real-time tasks.

**Weaknesses:**

1.	The model’s design is primarily an adaptation of established concepts from Vision Transformers (ViTs) and gated transformers, without substantial innovation in model architecture or theoretical insight. This limits the paper’s contribution beyond incremental improvements.

2.	The authors should select a single architecture to compare against prior state-of-the-art methods for a fair assessment, as previous works typically present only one variation. Comparisons of different PredFormer architectures should instead be placed in the ablation study section.

3.	The best results for both recurrent-based and recurrent-free methods should be highlighted in bold. Some results from prior methods appear to outperform PredFormer, and this should be clearly indicated.

4.	I noticed that STCF [1] shows superior performance on the TaxiBJ dataset. Why was it not considered in the comparisons?

[1] STCF: Spatial-Temporal Contrasting for Fine-Grained Urban Flow Inference

**Questions:**

Hope the author can answer the questions in the weaknesses part.

---

> ### Author Response · Authors · 2024-11-17
> **Response to Reviewer t3Do-Q1, Q2**
>
> **Q1 The model’s design is primarily an adaptation of established concepts from Vision Transformers (ViTs) and gated transformers, without substantial innovation in model architecture or theoretical insight. This limits the paper’s contribution beyond incremental improvements.**
>
> **A1:** We show that transferring the existing module design into the new task should be considered novel if good performance can be achieved. For example, ViT (ICLR 2021) extended Transformer (NeurIPS 2017) from natural language processing to computer vision, retaining the same architecture but demonstrating its effectiveness in a new domain. Similarly, our work uses the ViT structure in spatiotemporal predictive learning, a task with significantly different requirements for image classification. While transformer-based designs have been explored in various fields, we successfully adapt and apply such design with accuracy and efficiency balance to the specific task of spatiotemporal predictive learning. Our method achieves state-of-the-art results while maintaining high computational efficiency, even in challenging real-world scenarios such as high-resolution datasets.
>
> Moreover, our work goes beyond current Video ViT models, such as TimesFormer (ICML 2021), ViViT (ICCV 2021), and TSViT (CVPR 2023). We provide new insights into why the design principles of these models, which are primarily optimized for video classification, may not be effective for our task. By addressing these gaps, our work highlights critical adaptations and advances necessary for spatiotemporal predictive learning, establishing a considerable baseline for future research in this area.
>
> **Q2 The authors should select a single architecture and compare different PredFormer architectures in the ablation study.**
>
> **A2:** PredFormer variants represent task- and dataset-specific configurations, similar to the hyperparameter tuning typically performed in prior works. For instance, in SimVP, spatial and temporal hidden dimensions, spatial blocks, and temporal blocks are elaborately designed for each dataset to achieve optimal performance. Likewise, recurrent-based methods require extensive model hyperparameter tuning for different tasks. However, these final results are often reported under a single model name that represents their best configuration. One of our contributions is designing a pure transformer framework with PredFormer Encoder, and the presented variants demonstrate its adaptability, reflecting the tuning necessary to address specific task demands within this framework.
>
> Task and dataset characteristics further influence the choice of variant as explained in Line 296–298 and Table 1. Videos typically have shorter temporal intervals and longer temporal lengths, while real-world spatiotemporal data (e.g., TaxiBJ and WeatherBench) feature longer temporal intervals (e.g., 30 minutes for TaxiBJ and 1 hour for WeatherBench, as shown in Table 1) and varying spatial resolutions. These differences lead to distinct spatial-temporal dependencies, necessitating tailored designs for optimal performance. Placing these variants in an ablation study would not yield a definitive conclusion, as their effectiveness is inherently context-dependent.
>
> Finally, TSST is recommended as the starting configuration for achieving optimal performance across benchmarks, as detailed in Line 515–525. TSST consistently achieves state-of-the-art results and serves as a robust foundation for exploring other configurations.

---

> ### Author Response · Authors · 2024-11-17
> **Response to Reviewer t3Do-Q3, Q4**
>
> **Q3 The best results for both recurrent-based and recurrent-free methods should be highlighted in bold. Some results from prior methods appear to outperform PredFormer, and this should be clearly indicated.**
>
> **A3**: We appreciate the reviewer’s suggestion to highlight the best results for both recurrent-based and recurrent-free methods. We will revise the presentation to ensure that the best-performing results are appropriately emphasized in bold for clarity.
>
> In our paper, we show that PredFormer achieves state-of-the-art performance across all datasets, surpassing all baselines presented in this paper. If there are specific methods that "appear to outperform" PredFormer, we kindly ask the reviewer to indicate which table requires further clarification. Precise references of methods and benchmark datasets would allow us to provide a more focused and comprehensive response.
>
>
> **Q4 I noticed that STCF [1] shows superior performance on the TaxiBJ dataset. Why was it not considered in the comparisons?**
>
> **A4**: The main reason is that STCF and ours are in different experimental settings as follows:
>
> - **Dataset split**: PredFormer, along with the baselines we adopt, uses 20,461 samples for training and 500 samples for testing (as shown in Paper Table 1). In contrast, STCF uses 14,355 samples in total, with a 50%/25%/25% split for training, validation, and testing.
>  - **Spatial resolution**: PredFormer and the baselines use a 32×32 spatial resolution feature map for training and testing, while STCF uses a downsampled 32 ×
> 32 coarse-grained map to infer the 128 × 128 fine-grained map.
>  - **Training strategy**: PredFormer employs an end-to-end training approach, while STCF adopts a pretrain-finetune manner.
>
> For fair comparisons, we follow the settings of the previous benchmark OpenSTL (NeurIPS 2023). We will compare STCF with a similar setting in our paper in the future.

---

> ### Author Response · Authors · 2024-11-22
> **Follow-Up on Rebuttal Discussion**
>
> Dear Reviewer t3Do,
>
> We deeply appreciate your valuable feedback during the first round of review and the thoughtful discussion that has significantly helped us refine our work. We hope that our responses have addressed your concerns.
>
> We sincerely appreciate the opportunity to refine our work and kindly hope you might reconsider your concerns in light of our detailed responses and clarifications. Your insights are invaluable to us, and we would greatly welcome any additional feedback or suggestions you may have. Thank you once again for your time and thoughtful consideration.
>
> Looking forward to your reply.
>
> Best regards,
>
> Paper2123 Authors

---

> ### Author Response · Authors · 2024-11-24
> **Looking forward to your reply**
>
> Dear Reviewer t3Do,
>
> Thank you very much again for the time and effort put into reviewing our paper. We believe that we have addressed all your concerns in our response. We have also followed your suggestion to improve our paper and have added additional experimental analysis. We kindly remind you that we are approaching the end of the discussion period. We would love to know if there is any further concern, additional experiments, suggestions, or feedback, as we hope to have a chance to reply before the discussion phase ends.
>
> Best regards,
>
> Paper2123 Authors

---

> ### Author Response · Authors · 2024-11-27
> **Please let us know whether all the questions have been addressed**
>
> Dear Reviewer t3Do,
>
> As the deadline for the discussion phase will end soon, please let us know whether we have addressed all the questions.
>
> Thank you.
>
> Best regards,
>
> Paper2123 Authors

---

> > ### Comment · Reviewer_t3Do · 2024-11-30
> >
> > Considering that this year is ICLR 2025, the ViT architecture, originally introduced in ICLR 2021, is no longer novel, especially given its prior applications in spatio-temporal tasks. While the authors provide evidence of adapting ViT for their specific domain, this adaptation does not introduce significant innovation or novel modules tailored to spatio-temporal predictive tasks. Similar spatial and temporal attention block combinations have been extensively explored in the video prediction domain. I recommend developing new modules specifically tailored for spatio-temporal prediction to address domain-specific challenges. Additionally, refine the experiments to better highlight the benefits of these innovations. For now, I maintain my current score.

---

### Official Review · Reviewer_vtP1 · 2024-11-04

**Soundness:** 2
**Presentation:** 3
**Contribution:** 2
**Rating:** 5
**Confidence:** 4

**Summary:**

The paper presents PredFormer, a pure transformer-based model for spatiotemporal predictive learning, designed to address challenges in balancing computation costs with predictive accuracy. PredFormer leverages Gated Transformer Blocks (GTBs) and a comprehensive exploration of attention mechanisms to surpass previous recurrent-based and convolutional encoder-decoder methods.

**Strengths:**

- It boasts significant improvements in benchmark tasks involving datasets like Moving MNIST, TaxiBJ, and WeatherBench, showcasing both high accuracy and efficiency.
- The figures in the paper are clean and aesthetically pleasing, which enhances readability.

**Weaknesses:**

- Emphasizing "the first pure transformer model in xxx domain" as a main contribution feels minor, given that ViT was introduced in ICLR 2021. Highlighting this three years later may not meet ICLR's innovation standards.
  - Noting that SwinLSTM[1] and OpenSTL[2] have already shown the capability of Transformer in this domain, which will further reduce the novelty of this work.
  - Especially the OpenSTL, which already incorporates and evaluates commonly used transformer or CNN blocks, i.e., ViT, Swin Transformer, Uniformer, MLP-Mixer, ConvMixer, Poolformer, ConvNeXt, VAN, HorNet, and MogaNet.

- The study of spatiotemporal block combinations has already been extensively explored in `various domains` by `numerous Transformer-based works`, such as **Latte** [3]. Therefore, the contribution in this aspect may appear somewhat weak or incremental.
  - Due to the long sequence length problem of the Spatial-Temporal modeling task, most work delved into the architecture design of spatial block and temporal block, i.e., Latte also proposes 4 different spatial-temporal block variants, which are similar to this paper.

- The regularization and position encoding techniques used in this paper are well discovered in many language and vision papers. Utilizing them directly provides limited contribution.


[1] SwinLSTM: Improving Spatiotemporal Prediction Accuracy using Swin Transformer and LSTM

[2] OpenSTL: A Comprehensive Benchmark of Spatio-Temporal Predictive Learning

[3] Latte: Latent diffusion transformer for video generation

**Questions:**

Please refer to the weakness part. I will raise my scores if the authors can resolve my concerns.

---

> ### Author Response · Authors · 2024-11-17
> **Response to Reviewer vtP1-Q1**
>
> **Q1: SwinLSTM and OpenSTL further reduce the novelty of this work, as OpenSTL already integrates transformer and CNN blocks.**
>
> **A1:** Rather than reducing the novelty of our work, SwinLSTM and OpenSTL actually serve as important motivations for this work and further highlight our contributions. As stated in Lines 51–74 and Figure 1 of our main paper, the key differences are as follows:
> - SwinLSTM is based on a recurrent framework (originating from ConvLSTM), which is inherently limited by its recurrent processing. This leads to slower training speeds, higher GPU memory consumption, and lower performance. Instead, our model is recurrent-free and thus achieves significantly higher efficiency.
> - OpenSTL is a CNN-based encoder-decoder framework that employs convolution kernels and attention modules for spatial and temporal modeling. However, the performance of CNN design is inherently limited by their local receptive fields and inductive biases. In contrast, our model adopts a pure Transformer architecture, offering a global receptive field and delivering significant performance improvements.
>
> We present extensive experimental results on three datasets in Table A1, A2, A3, and A4 to support our claims, where the training epochs are the same as  SwinLSTM and OpenSTL for fair comparisons.
>
>
> Table A1. Comparisons of PredFormer and SwinLSTM on the Moving MNIST dataset with 2000 training epochs.
> | Method         | Par**as(M)** | **Flops(G)** | **Training Epoch Time** | **MSE**  | **SSIM**  |
> | -------------- | ------------ | ------------ | -------------- | -------- | --------- |
> | SwinLSTM       | --           | --           | 9min           | 17.7     | 0.962     |
> | PredFormer | 25.3         | 16.5         | **3.5min**         | **12.5** | **0.973** |
>
>
> Table A2. Comparisons of PredFormer and OpenSTL on the Moving MNIST dataset with 200 training epochs.
> | Method                   | Paras(M) | Flops(G) | MSE      | MAE      | SSIM      |
> | ------------------------ | -------- | -------- | -------- | -------- | --------- |
> | OpenSTL+ViT              | 46.1     | 16.9     | 35.2     | 95.9     | 0.914     |
> | OpenSTL+Swin Transformer | 46.1     | 16.4     | 29.7     | 84.1     | 0.933     |
> | PredFormer         | **25.3** | **16.5** | **26.0** | **77.2** | **0.941** |
>
>
> Table A3. Comparisons of PredFormer, SwinLSTM, and OpenSTL on the TaxiBJ dataset.
> |     Method          | Par**as(M)** | **Flops(G)** | **FPS** | **MSE**  | **SSIM**  |
> | -------------- | ------------ | ------------ | -------------- | -------- | --------- |
> | SwinLSTM       | **2.9**           | **1.3**           | 1425           | 0.303     | 0.984     |
> | OpenSTL+ViT              | 9.7      | 2.8      | 1301     | 0.317  | 0.984     |
> | OpenSTL+Swin Transformer | 9.7      | 2.6      | 1506     | 0.313  | 0.984     |
> | PredFormer | 6.3         | 1.6         | **2364**     | **0.277** | **0.986** |
>
> Table A4. Comparisons of PredFormer and OpenSTL on the Human3.6m dataset.
> |              | Paras(M) | Flops(G) | FPS  | MSE       | MAE        |
> | ------------ | -------- | -------- | ---- | --------- | ---------- |
> | OpenSTL+ViT  | 28.3     | 239.0    | 17   | 136.3     | 1603.5     |
> | OpenSTL+Swin Transformer | 38.8     | 188.0    | 28   | 133.2     | 1509.7     |
> | PredFormer   | **12.7**     | **65.2**     | **78**   | **114.7**     | **1403.6**     |

---

> ### Author Response · Authors · 2024-11-17
> **Response to Reviewer vtP1-Q2, Q3, Q4**
>
> **Q2: Emphasizing "the first pure transformer model in xxx domain" feels minor.**
>
> **A2:** We have removed this sentence in our revised paper to avoid overemphasizing the point.
>
> **Q3 Previous work also explored spatiotemporal block combinations, such as Latte, which weaken our contribution.**
>
> **A3:** Spatial-temporal factorization is an essential module for processing spatio-temporal sequences. We acknowledge some shared designs between Latte and our work. For example, variants a and b in Latte are similar to the Binary-ST and Fac-ST modules designed in PredFormer. Additionally, variants b and d in Latte utilize shared MLPs for both spatial and temporal self-attention. Except for the four variants in Latte, we conduct a more systematic analysis of spatial-temporal design such as full attention, Fac-TS, and six interleaved variants.
>
> However, its performance varies depending on the specific tasks. While Latte focuses on video generation, our work is for spatial-temporal predictive learning. Therefore, our key conclusion is novel and distinct from Latte due to the differences in the two tasks. Specifically, we find that Fac-ST with the spatial-first design performs worse than Fac-TS with the temporal-first design on our task. This discrepancy may be attributed to the data characteristics used in the tasks. Videos used in video generation typically have shorter temporal intervals and longer temporal lengths, while real-world spatiotemporal data used in spatial-temporal predictive learning often have longer temporal intervals (e.g., 30 minutes for TaxiBJ, 1 hour for WeatherBench). These differences pose the greater importance of temporal-first designs for spatiotemporal predictive learning.
>
> As suggested, we have cited Latte in our revised paper to strengthen our discussion.
>
> **Q4: The regularization and position encoding techniques are well discovered. Utilizing them directly provides limited contribution.**
>
> **A4:** The strength of PredFormer lies in its careful selection of well-established techniques, such as regularization and positional encoding, validated through numerous ablation studies.
>
> - **Regularization**: Mainstream regularization methods, such as those used in SimVP and OpenSTL, typically employ scaled drop path strategies. However, as demonstrated in our ablation study (Table 7), using a scaled drop path alone is insufficient for PredFormer to achieve significantly better performance than SimVP and TAU.
> - **Positional Encoding**: Leading video classification methods, such as ViViT and TSViT, rely on learnable positional embeddings. However, as shown in our ablation studies (Table 7), these embeddings are less effective for spatiotemporal predictive learning.
>
> We will include these explanations in the corresponding ablation study section of our main paper.

---

> ### Author Response · Authors · 2024-11-21
> **Follow-Up on Rebuttal Discussion**
>
> Dear Reviewer vtP1,
>
> We deeply appreciate your valuable feedback during the first round of review and the thoughtful discussion that has significantly helped us refine our work. We hope that our responses have addressed your concerns.
>
> We sincerely appreciate the opportunity to refine our work and kindly hope you might reconsider your concerns in light of our detailed responses and clarifications. Your insights are invaluable to us, and we would greatly welcome any additional feedback or suggestions you may have. Thank you once again for your time and thoughtful consideration.
>
> Looking forward to your reply.
>
> Best regards,
>
> Paper2123 Authors

---

> ### Author Response · Authors · 2024-11-24
> **Looking forward to your reply**
>
> Dear Reviewer vtP1,
>
> Thank you very much again for the time and effort put into reviewing our paper. We believe that we have addressed all your concerns in our response. We have also followed your suggestion to improve our paper and have added additional experimental analysis. We kindly remind you that we are approaching the end of the discussion period. We would love to know if there is any further concern, additional experiments, suggestions, or feedback, as we hope to have a chance to reply before the discussion phase ends.
>
> Best regards,
>
> Paper2123 Authors

---

> ### Comment · Reviewer_vtP1 · 2024-11-24
>
> I have read all the rebuttals from the authors, and I still have a concern:
>
> - `However, the performance of CNN design is inherently limited by their local receptive fields and inductive biases.`
>   - Is there some theoretical evidence or analysis to support this viewpoint, i.e., local receptive fields and inductive biases limit the performance of this task?

---

> ### Author Response · Authors · 2024-11-25
> **Response to Reviewer vtP1-Q5**
>
> **Q5: 'However, the performance of CNN design is inherently limited by their local receptive fields and inductive biases.' Is there some theoretical evidence or analysis to support this viewpoint, i.e., local receptive fields and inductive biases limit the performance of this task?**
>
> **A5:** This statement is supported by Effective Receptive Field (ERF) analysis [1] and prior research.
>
> The constraints imposed by local receptive fields and inductive biases in convolutional neural networks (CNNs) have been widely analyzed in computer vision tasks. ERF  (NeurIPS16) [1] shows that the effective receptive field in CNNs is significantly smaller than the theoretical receptive field, limiting their ability to model global context effectively. Existing CNN architectures, such as VGGNet, rely on stacking small 3×3 convolutional kernels, which restrict the receptive field and limit global context modeling. This design principle is also prevalent in spatiotemporal predictive learning models like SimVP, TAU, and OpenSTL, which rely on 3×3 convolution kernels in their encoder-decoder structures.
>
> We note previous studies have demonstrated that increasing kernel sizes can overcome the limitations of local receptive fields. For example, RepLKNet (CVPR 2022) [2] expands kernels to sizes as large as 31×31, significantly increasing the ERF and improving performance by better capturing global spatial dependencies. Similarly, ConvNeXt (CVPR 2022) [3] adopts larger kernel sizes to reduce reliance on local receptive fields, enabling broader spatial coverage and achieving performance comparable to transformer-based architectures. These studies highlight how existing CNN designs, with their reliance on small kernels, inherently limit the effective receptive field and, consequently, their ability to capture global context. While these works focus on improving CNNs by enlarging kernels to match the global receptive field of transformers, they also emphasize the inherent limitations of conventional CNNs, such as those used in SimVP, TAU, and OpenSTL.
>
> For the spatiotemporal predictive tasks, capturing both spatial and temporal dependencies is critical. If CNNs are constrained in their ability to model global spatial relationships due to local receptive fields, it logically follows that these limitations extend to the joint modeling of spatial and temporal dependencies. Consequently, transformers, with their global attention mechanisms, are better suited for spatiotemporal tasks, as they can effectively capture long-range dependencies across both dimensions.
>
> In addition, SimVP and TAU demonstrate the effectiveness of expanding receptive fields. SimVP employs CNNs with 3×3 kernels for spatial modeling and employs Inception-Unet for temporal modeling. TAU, building on SimVP with 3×3 kernels for spatial modeling, incorporates a DW-DW D-1×1 module as its temporal module, effectively simulating larger convolutional kernels. This approach improves spatiotemporal predictive learning by addressing the limited temporal receptive field, but it still models spatial relations using 3×3 kernels, constrained by local receptive fields.
>
> Inspired by these works, PredFormer takes the next step by replacing CNNs with transformers in both spatial and temporal dimensions. This eliminates the constraints of local receptive fields and inductive biases, offering a comprehensive solution for capturing global context in spatiotemporal predictive learning tasks.
>
> Extensive experimental results demonstrate the contributions of PredFormer. As discussed in Table 2 and Table 3 of this paper, when training is extended to 2000 epochs, PredFormer achieves a higher upper bound compared to CNN-based methods such as SimVP and TAU. Moreover, PredFormer not only surpasses these methods in performance but also achieves this with significantly fewer parameters across multiple benchmarks.
>
>
>
> **References:**
>
> [1] Understanding the Effective Receptive Field in Deep Convolutional Neural Networks.
>
> [2] Scaling Up Your Kernels to 31×31: Revisiting Large Kernel Design in CNNs.
>
> [3] A ConvNet for the 2020s.

---

> ### Comment · Reviewer_vtP1 · 2024-11-25
>
> Agree with Reviewer sE64. Although this paper shows great efficiency and effectiveness in the spatial-temporal domain, I can not learn something interesting or insightful from it. The excellent performance and fluent writing make this article convincing, but there is still some distance from the novelty standard of ICLR to some extent.
>
> Cause I am not an indeed expert in this domain, I would increase the final score if there were some reviewers gave more guarantee of this article's importance in this domain.

---

### Author Response · Authors · 2024-11-17
**Overall Response**

## **Overall Response:**

We thank the reviewers for acknowledging the strengths of our paper:

- Well-written content, clean figures, and easy-to-follow explanations.
- Significant performance improvements with high efficiency across multiple datasets.

We appreciate the constructive feedback from all reviewers and add additional experiments and more precise explanations of our contributions in the revised paper.

First, we address common questions and provide detailed responses to the questions posed by each reviewer.

### Novelty and Contribution (R#t3Do, R#sE64)

Compared to previous works based on RNN and CNN architectures, PredFormer introduces a simple yet effective pure-transformer framework for spatial-temporal predictive learning. While the transformer-based designs have been explored in various domains, we present the **meticulously designed gated transformer blocks that accommodate full-, factorized-, and interleaved- spatial-temporal attention**, that helps **achieving state-of-the-art results with high efficiency**, even in challenging real-world scenarios such as high-resolution datasets. Furthermore, our work provides comprehensive insights into spatiotemporal factorization, gated transformer block, positional encoding, and regularization techniques, offering advancements beyond prior Video ViT methods. Our work can serve as a strong baseline to promote spatial-temporal predictive learning. We would like to remind reviewers that this work is developed based on the  "*simplicity is the ultimate sophistication*" principle, and achieves state-of-the-art performance.

### Task Difference between Video Prediction and Generation (R#vtP1, R#sE64)
- Video prediction and video generation differ in their goals and data characteristics. Video prediction emphasizes pixel-level accuracy, aiming to capture spatiotemporal relationships precisely and adhere to physical-world rules. In contrast, video generation focuses on unseen objects and scenes and the results are typically evaluated based on perceptual quality and consistency using metrics such as FVD. Additionally, videos typically have shorter temporal intervals and longer sequences, while real-world spatiotemporal data, such as traffic or weather datasets, often feature longer intervals and shorter temporal lengths. These distinctions result in different preferences for factorization approaches in the respective domains.
- Compared with Latte (a recent latent diffusion transformer for video generation), PredFormer introduces an effective spatiotemporal factorization method that not only differs from but also extends beyond prior Video ViT designs. Furthermore, the goals and findings of this work are significantly different from Latte, due to the specific characteristics of spatiotemporal predictive learning tasks, which differ from video generation. For example, we find that Fac-ST (spatial-first) performs worse than Fac-TS (temporal-first) for spatiotemporal predictive learning.
- PredFormer, along with SimVP and PredRNN, belongs to the deterministic model category, which directly predicts pixel values. Deterministic models aim for precision and efficiency in spatiotemporal predictions. On the other hand, probabilistic models, such as GANs, VAEs, and diffusion models, operate in latent spaces and are designed to generate diverse outputs by sampling from probability distributions. These models often trade off inference speed for greater flexibility and diversity in generated results.

---

### Note · Authors · 2025-02-15

I have read and agree with the venue's withdrawal policy on behalf of myself and my co-authors.

---

### Meta-Review · Area_Chair_hQ9N · 2024-12-18

**Metareview:**

The paper introduces PredFormer, a transformer-based model for spatiotemporal predictive learning, aiming to replace traditional recurrent and convolutional architectures with Gated Transformer Blocks (GTBs). The model leverages various attention mechanisms to capture complex spatial and temporal relationships, achieving strong results on datasets like Moving MNIST, TaxiBJ, and WeatherBench.

The strengths of the paper include performance improvements in accuracy and efficiency, as highlighted by Reviewers e793 and t3Do. The model is evaluated on multiple datasets, demonstrating adaptability across different spatiotemporal tasks, as noted by Reviewer t3Do. The paper is well-written and organized, making it easy to follow, as mentioned by Reviewer sE64. Additionally, the extensive experiments and ablation studies highlight the advantages of the proposed model, and there are plans to release the source code and trained models, facilitating further research and practical applications.

However, the paper has several weaknesses. Reviewer vtP1 points out the limited novelty, as the model primarily adapts existing transformer concepts without substantial innovation. Reviewer e793 notes that the model has not been evaluated on more complex datasets like Traffic4Cast or Human3.6M, limiting its validation scope. Reviewer t3Do highlights the lack of comparison with some state-of-the-art methods, such as Earthformer, which limits the comprehensiveness of the evaluation. Additionally, PredFormer does not demonstrate a significant advantage in running efficiency compared to recurrent-free methods, and its performance in terms of MSE is similar, as noted by Reviewer e793. The contribution is seen as incremental, with the "first pure transformer model" claim feeling minor, as mentioned by Reviewer vtP1.

The AC aligns with the reject recommendation of all reviewers. Some concerns raised by the reviewers include that the contributions are primarily incremental, with limited innovation in model architecture or theoretical insights. The evaluation is insufficient, lacking complex datasets and comprehensive comparisons with state-of-the-art methods. The claimed improvements in efficiency and performance are not compelling enough, and the overall impact does not meet ICLR's acceptance threshold. The AC highly recommends the authors to address the concerns of the reviewers and take into account their suggestions of improvement when preparing a revised version. Especially the extra experiments promised in the rebuttal are crucial for justifying the method and could be included before resubmitting the work.

**Additional Comments On Reviewer Discussion:**

During the rebuttal period, the authors addressed several points raised by the reviewers, but the responses did not significantly alter the reviewers' initial assessments.

Reviewer vtP1 acknowledged the paper's efficiency and effectiveness in the spatiotemporal domain but maintained concerns about its novelty. The reviewer expressed willingness to increase the score if other reviewers could guarantee the paper's importance in the domain, but no such assurances were provided.

Reviewer t3Do reiterated that the ViT architecture, introduced in ICLR 2021, is no longer novel, especially given its prior applications in spatiotemporal tasks. The reviewer suggested that the authors develop new modules specifically tailored for spatiotemporal prediction to address domain-specific challenges. The authors' rebuttal did not sufficiently demonstrate significant innovation or novel contributions, leading the reviewer to maintain their score.

Reviewer sE64 found the paper's contributions to be marginal, describing the architectural designs as task-specific ablations of existing techniques rather than fundamentally novel ideas. The reviewer also disagreed with the authors' claim that deterministic models are inherently more precise than probabilistic models, noting that generative models can offer greater precision by modeling the distribution of spatiotemporal data more finely. The rebuttal did not address these concerns to the reviewer's satisfaction, resulting in the maintenance of the original score.

In weighing these points for the final decision, the primary concern remained the lack of significant novelty and innovation in the paper. While the authors provided clarifications and additional context, the reviewers' concerns about the paper's contributions being incremental and not meeting the novelty standards of ICLR were not sufficiently addressed. Consequently, the decision to reject the paper was upheld.

---

### Decision · Program_Chairs · 2025-01-22

Reject